# Synergistic aluminum dual-atom sites and nickel nanoclusters for acetylene selective hydrogenation

Yanan Liu [1,2,5], He Yu[1,5], Mengjiao Li[3,5], Li Yan[1,5], Ruihu Lu [3], Xiuting Fu[1], Zhenfei Zhang[1], Youqi Zhu[4], Ziyun Wang [3] ✉ & Shubo Tian [1,2] ✉

Synergistic catalysis, where distinct active species collaboratively activate different substrates, provides a powerful strategy for achieving chemical transformations with enhanced efficiency. Although $Al_2O_3$ and bulk aluminum species are widely employed as catalyst supports, they are seldom regarded as active centers, especially in hydrogenation. Here, we show that atomically dispersed Al species can catalyze acetylene conversion at elevated temperatures. Building on this insight, we have designed a synergistic catalyst featuring precisely controlled Al dual-atom sites paired with Ni nanoclusters, synthesized via a solid-transformation-coupled gas-adsorption strategy to overcome the typical activity-selectivity trade-off. Under mild, cost-effective conditions, this catalyst achieves nearly full acetylene conversion with ~90% ethylene selectivity and excellent long-term stability. In situ spectroscopy and theoretical calculations reveal a cooperative mechanism: Ni nanoclusters efficiently dissociate $H_2$ into active hydrogen species (H*), while adjacent Al dual-atom sites shuttle the H* species to π-adsorbed acetylene, lowering the energy barrier for ethylene formation compared to over-hydrogenation and coke formation.

Selective acetylene hydrogenation is a critical process in the modern polyolefin industry upgrading for ethylene purification[1,2], as it removes approximately 1% acetylene impurities generated mainly by catalytic cracking[3,4]. Although supported Pd-based catalysts have achieved significant industrial success[5–7], their practical application is limited by the suboptimal activity/selectivity trade-offs and scarcity of precious metals[8,9]. This drives exploration of supported non-noble metal alternatives that offer cost-effectiveness and elemental abundance, though these systems often suffer from thermodynamically favored over-hydrogenation of ethylene and undesired polymerization on metal surface[10,11]. Meanwhile, conventional $Al_2O_3$ supports, while widely utilized for their low cost, mechanical robustness, and chemical stability,

exhibit the limitation in enhancing catalytic performance as active species. Recent advances in support engineering through atomic-level modifications have emerged as promising strategies for catalyst optimization[12–15]. Despite their ability to unlock novel reaction pathways and deliver exceptional catalytic performance—thanks to unique Lewis acidity modulation and surface-reconstruction effects—atomically dispersed aluminum sites remain largely unexplored.

Recent research advancements have demonstrated that atomically precise catalysts, particularly single-atom catalysts (SAC) and dual-atom catalysts (DAC), have emerged as promising candidates for various reactions, due to their precisely tunable electronic and geometric properties[16–18], diverse configurations of adsorbed species[19,20],

[1]State Key Laboratory of Chemical Resource Engineering, Beijing Engineering Center for Hierarchical Catalysts, Beijing University of Chemical Technology, Beijing, China. [2]Quzhou Institute for Innovation in Resource Chemical Engineering, Quzhou, China. [3]School of Chemical Sciences, University of Auckland, Auckland, New Zealand. [4]Beijing Key Laboratory of Construction Tailorable Advanced Functional Materials and Green Applications, School of Materials Science and Engineering, Beijing Institute of Technology, Beijing, China. [5]These authors contributed equally: Yanan Liu, He Yu, Mengjiao Li, Li Yan. ✉e-mail: ziyun.wang@auckland.ac.nz; tianshubo@mail.buct.edu.cn

and resulted unique performances[21,22]. Nevertheless, atomically dispersed sites often exhibit lower catalytic activity compared to their nanoparticle counterparts in scenarios where multiple active sites are required[23]. Therefore, further optimization of active sites is essential to achieve both high catalytic selectivity and activity. Synergistic catalysis can facilitate reactions between two activated intermediates, driven by distinct metal active sites acting on different substrates, thereby enhancing catalytic performance[24–26]. Based on this, some researchers have proposed efficient synergistic catalysis systems integrating the SAC and the metal nanoparticles/nanoclusters (NPs/NCs). This demonstrates the simultaneous integration and regulation of catalytic activity and selectivity, compensating for shortcomings of one-component catalysts[27–29]. However, research on synergistic catalysts that combine more flexible dual-atom sites with other types of catalytic sites is scarce, like the dual-atom and nanocluster/nanoparticle synergistic system, and the controllable fabrication methods for this system are still in their infancy. Moreover, despite the potential of synergistic catalysis, its application in complex reactions remains limited due to challenges such as undesired mono-catalytic pathways generating side products and the deactivation by metal detachment or aggregation[30].

Inspired by the above insight, we were initially surprised to discover that, unlike the inert $Al_2O_3$ support, atomic-level Al sites exhibit the ability to selectively activate acetylene into ethylene, despite their limited capacity for $H_2$ dissociation. To enhance both selectivity and activity, we developed a two-step synthesis strategy, solid transformation coupled with gas adsorption, to construct the synergistic catalytic sites comprising Al dual-atom sites and Ni nanoclusters ($Al_2$-$Ni_{NC}$/NCNT). This catalyst achieves up to 90% ethylene selectivity at nearly full conversion in selective acetylene hydrogenation, even under front-end ($H_2$-rich) reaction conditions, and no visible activity decay after 100 h of long-term testing. Our findings reveal that concerted $Al_2$ decorating with Ni nanoclusters efficiently facilitates cleavage of C ≡ C and H-H bonds while retaining the C = C bond. Through in situ characterization and theoretical calculations, we demonstrate that Ni nanoclusters enhance the hydrogen activation, whereas Al dual-atom drives acetylene adsorption, conversion, and active hydrogen transfer. Additionally, this work not only provides a strategy to overcome the trade-off between activity and selectivity and stability in Ni-based catalysts but also establishes a novel concerted catalysis integrating dual-atom sites with nanoclusters.

## Results
### Catalyst design and structural characterization
For the first time, we synthesized an Al dual-atom catalyst via a sublimation transformation strategy to explore the potential of atomic-level Al catalysts in the selective hydrogenation of acetylene. As shown in Fig. 1a, anhydrous aluminum chloride ($AlCl_3$) and nitrogen-doped carbon nanotube (NCNT) were sequentially placed in two porcelain boats at 180 °C under an argon atmosphere. This setup facilitated the generation of $Al_2Cl_6$ vapor from the sublimation of $AlCl_3$, which was subsequently captured by NCNT. The resulting $Al_2Cl_6$/NCNT composite was then collected and heated to 550 °C under a 5% hydrogen atmosphere to remove Cl ligands from the Al sites. This process effectively stabilized the Al dimers and facilitated their bonding with the N sites of the NCNT, resulting in the formation of a well-defined Al dual-atom catalyst ($Al_2$/NCNT). With the goal of improving practical utility, $Al_2$-$Ni_{NC}$/NCNT synergistic catalysts were further fabricated through a gas adsorption method. In this process, the pre-synthesized $Al_2$/NCNT was exposed to a $NiCl_2$ vapor stream for deposition, leading to the formation of Ni nanoclusters surrounded by $Al_2$ dual-atom sites. Meanwhile, the catalysts composed exclusively of Al single-atom (denoted as $Al_1$/NCNT) and Ni nanocluster (denoted as $Ni_{NC}$/NCNT) were also prepared for comparison. Inductively coupled plasma-optical emission spectroscopy (ICP-OES, Supplementary Table 1)

reveals that Ni and Al contents in $Al_2$-$Ni_{NC}$/NCNT are 0.28 wt.% and 0.43 wt.%, comparable to Ni nanoclusters and Al single-atom counterparts, respectively. In addition to the diffraction features from the NCNT support, the X-ray diffraction (XRD) pattern of $Al_2$-$Ni_{NC}$/NCNT exhibits the weak characteristic diffraction peaks at 44.5° and 51.8° (PDF#04-0850, Fig. 1b), similar to those observed in $Ni_{NC}$/NCNT. In contrast, no diffraction signals attributable to Al species are detected in the $Al_2$/NCNT and $Al_1$/NCNT samples (Fig. 1b). As shown in Supplementary Figs. 1–3, all catalysts maintain the characteristic tubular morphology of NCNT, with metallic nanoclusters uniformly confined within the NCNT support in $Al_2$-$Ni_{NC}$/NCNT, and $Ni_{NC}$/NCNT samples. Notably, the support properties remain unchanged, as evidenced by Raman spectra (Supplementary Fig. 4), in which the spectra display similar patterns with the characteristic D peak at 1338 $cm^{-1}$ and G peak at 1562 $cm^{-1}$, along with comparable $I_D/I_G$ intensity ratios across $Al_2$-$Ni_{NC}$/NCNT, $Al_2$/NCNT and $Ni_{NC}$/NCNT catalysts. This consistency indicates that the carbon structures exhibit similar levels of disorder or defects. Meanwhile, the metallic Ni nanoclusters and Al dual-atom species shows no obvious Raman signals primarily due to their high polarizability, which hinders atomic vibrations from inducing measurable changes in polarizability[31]. Furthermore, aberration-corrected high-angle annular dark-field scanning transmission electron microscopy (AC HAADF-STEM) imaging reveals that the dispersed Al single-atom and dual-atom sites in $Al_1$/NCNT and $Al_2$/NCNT are clearly observed on the NCNT support, highlighted by white circles in Fig. 1c, e and Supplementary Fig. 5. Following Ni deposition, as shown in Fig. 1d and Supplementary Fig. 5d, small nanoclusters with the lattice distances of 0.206 nm (corresponding to the Ni (111) plane) are observed, indicating a crystalline Ni structure with an average particle size of approximately 2.5 nm. Additionally, it is observed that while the Al distribution remains uniform, the Ni nanoclusters are segregated and surrounded by abundant Al dual-atom in $Al_2$-$Ni_{NC}$/NCNT, as confirmed by the HAADF-STEM and EDS image (Fig. 1d and Supplementary Fig. 6).

X-ray absorption spectra (XAS) analysis was performed to elucidate the electronic and coordination structures of the active sites. As shown in Fig. 2a, the Al K-edge X-ray absorption near-edge structure (XANES) result reveals that the absorption edges of $Al_2$-$Ni_{NC}$/NCNT, $Al_1$/NCNT, and $Al_2$/NCNT are positioned between $Al_2O_3$ and Al foil, indicating that the average oxidation state of Al species is likely greater than 0 but less than +3. Moreover, compared to $Al_1$/NCNT and $Al_2$/NCNT, the absorption edge of $Al_2$-$Ni_{NC}$/NCNT shifts to higher photon energies, indicating a higher electron deficiency for Al species in $Al_2$-$Ni_{NC}$/NCNT, implying higher oxidation states. This conclusion is supported by the Al $2p$ XPS results, which reveal peak shifts toward higher binding energies (Supplementary Fig. 7). Notably, the absorption edge of Ni species in both $Al_2$-$Ni_{NC}$/NCNT and $Ni_{NC}$/NCNT samples aligns closer with that of Ni foil than NiO, indicating that the Ni species predominantly exist in a metallic state. Consistently, the absorption edge in $Al_2$-$Ni_{NC}$/NCNT exhibits a negative shift (Fig. 2b), implying an increased electrons on Ni species, which suggests that there is an electron interaction between neighboring Ni nanocluster and Al dual-atom.

Based on this, Fourier-transform extended X-ray absorption fine structure (FT-EXAFS) was utilized to investigate the coordination environment of Al species (Fig. 2c). The two prominent peaks for $Al_2$-$Ni_{NC}$/NCNT and $Al_2$/NCNT appear at approximately 1.3 and 2.2 Å (without phase correction) corresponding to Al-N and Al-Al coordination, respectively[32], whereas $Al_1$/NCNT exhibits only the former feature. Notably, the possibility of Al-Cl or Al-O coordination is excluded due to the higher peak position of M-Cl coordination at ~1.9 Å[33], and the evidence of Al interaction with N species from XPS results (Supplementary Figs. 8, 9). More importantly, the second peaks in $Al_2$-$Ni_{NC}$/NCNT and $Al_2$/NCNT are both shorter than the typical Al-Al distance of ~2.4 Å observed in Al foil. Additionally, wavelet transforms analysis of Al EXAFS oscillations further corroborates the formation of

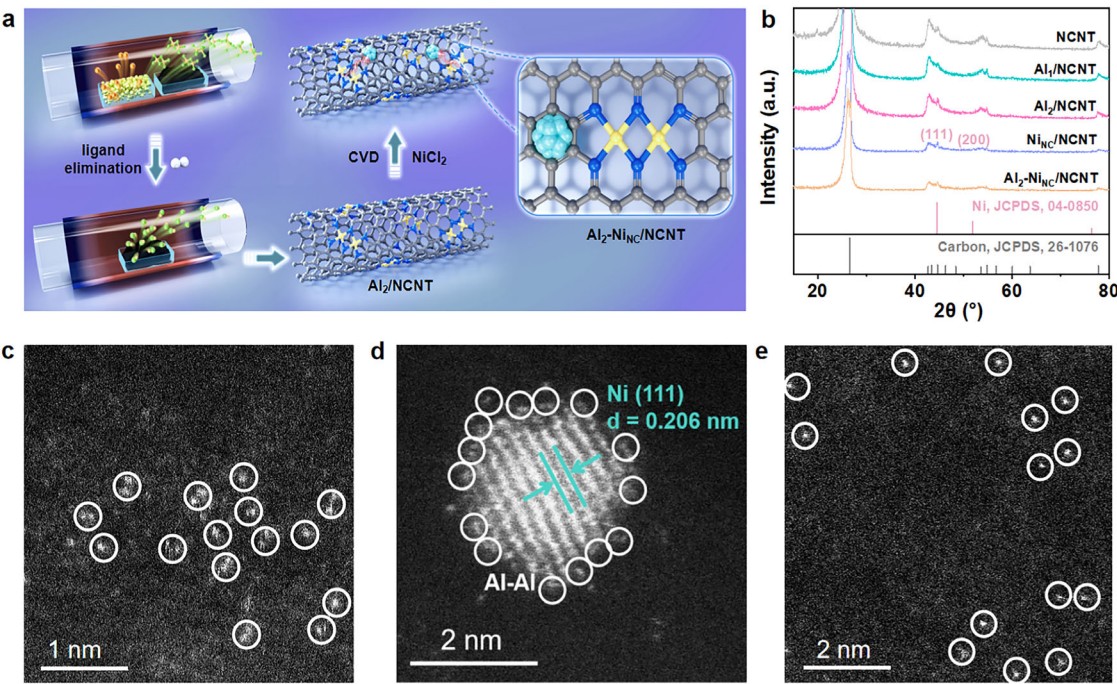

**Fig. 1 | Synthesis principles and catalytic properties. a** Schematic illustration of the synthesis of the Al₂-Ni_NC/NCNT synergistic catalyst (grey: C; blue: N; yellow: Al; light blue: Ni). **b** XRD patterns; Aberration-corrected HAADF-STEM images of (**c**) Al₂/NCNT (**d**) Al₂-Ni_NC/NCNT (**e**) Al₁/NCNT.

Al dual-atom sites (Fig. 2g). The WT-EXAFS contour plot of Al₂-Ni_NC/NCNT exhibits two intensity maxima at ~4.3 and ~4.9 Å⁻¹, attributed to the Al-Al and Al-N interaction pathways, respectively. To further quantify the coordination environment of Al atoms, EXAFS fittings were performed. As shown in Fig. 2e, Supplementary Figs. 10, 11 and Supplementary Table 2, the best-fit EXAFS result for Al₂-Ni_NC/NCNT indicates an average Al-N coordination number of 3.9 with a bond length of 1.78 Å, while the Al-Al coordination number is 0.9 with a bond length of 2.32 Å, highly consistent with the local atomic configuration of Al₂/NCNT. These findings strongly corroborate that Al species in Al₂-Ni_NC/NCNT exist as Al₂ dual-atom rather than binding directly to Ni.

In contrast, the Ni K-edge EXAFS spectra of Al₂-Ni_NC/NCNT exhibit a dominant peak at 2.1 Å with a shoulder at 1.4 Å (Fig. 2d), indicative of Ni-Ni and Ni-N coordination with a lower coordination number. This interpretation is reinforced by the EXAFS oscillation in k space, which exhibits shorter periods and weaker amplitudes compared to the referenced Ni catalyst and Ni foil, as confirmed by the wavelet transform spectra (Fig. 2h). EXAFS fittings (Fig. 2f, Supplementary Figs. 12, 13 and Supplementary Table 3) further reveal a reduction in coordination numbers for both Ni-Ni and Ni-N pairs, suggesting the formation of smaller Ni nanoclusters interacting with the NCNT support.

**Catalytic performance of Al₂-Ni_NC/NCNT**
The performances of acetylene hydrogenation in an excess ethylene stream were evaluated for all prepared catalysts with similar Al and Ni loadings using a fixed-bed reactor. The catalytic conversion and selectivity of Al₁- and Al₂- based samples supported on NCNT were first evaluated in acetylene selective hydrogenation, with the concentration of reactants and products as a function of temperature shown in Fig. 3. Remarkably, atomic-scale Al sites exhibit excellent activation ability of acetylene on the premise of excluding the interference of NCNT support (Supplementary Fig. 14) and reactor system (Supplementary Fig. 15), with Al₂/NCNT displaying better catalytic performance compared to single-atom Al counterpart (Fig. 3a). This behavior markedly contrasts with the inert α-Al₂O₃ and Al powders, which is usually used as substrate to load Pd catalysts for C₂ selective hydrogenation in the

industry. However, higher acetylene conversion for Al₂/NCNT catalysts requires the elevated reaction temperature (> 240 °C in Fig. 3a and Supplementary Fig. 16) although exhibiting excellent ethylene selectivity (94.0%). Moreover, the synergistic catalytic system comprising Ni nanoclusters with exceptional hydrogen dissociation and Al dual-atom sites for acetylene activation was further evaluated. Various flow rates, H₂/hydrocarbon ratio, and temperatures were systematically tested to determine the optimal reaction conditions (Fig. 3 and Supplementary Table 4). As expected, Al₂-Ni_NC/NCNT significantly enhances catalytic activity, achieving 99.98% conversion at 157 °C, far superior to Al₂/NCNT (49.75% @ 340 °C) and Ni_NC/NCNT (99.93% @ 200 °C), as shown in Fig. 3a and Supplementary Figs. 17, 18. More importantly, even at complete acetylene conversion, Al₂-Ni_NC/NCNT maintains ethylene selectivity of 90.26% (Fig. 3b), with only minor formation of ethane (6.17% of selectivity) and oligomers (3.57% of selectivity), as depicted in Fig. 3c-d. In contrast, the Ni_NC catalyst, which lacks adjacent Al dual-atom, exhibits markedly lower ethylene selectivity (43.70% at 99.93% conversion). Based on the temperature-dependent concentration profiles of acetylene, ethylene, ethane and oligomers (Supplementary Figs. 17, 18), it is found that ethane and oligomers start to form over Al₂-Ni_NC/NCNT catalyst when the temperature increases to 88 °C; however, their corresponding amounts are significantly lower than that of Ni_NC catalyst. This difference may be attributed to the formation of ethylidene (=CHCH₃) on Ni nanoclusters, resulting in excessive hydrogenation to ethane and polymerization into oligomers (Fig. 3c, d). Meanwhile, the lower formation rate of oligomers of 0.29 mol_oligomers mol_Ni⁻¹ h⁻¹ over Al₂-Ni_NC/NCNT in Supplementary Fig. 19, compared to that of Ni_NC/NCNT (1.72 mol_oligomers mol_Ni⁻¹ h⁻¹) indicates that the construction of synergistic sites could suppress the deposition of carbonaceous species. These findings demonstrate that synergy between Al dual-atom sites and adjacent Ni nanoclusters not only drives the different product distribution but also enhances the activity and selectivity. With lower metal loading and reduced reaction temperature, this system outperforms most reported non-noble metal catalysts and even rivals precious metal catalysts (Supplementary Tables 5–7)[34], making it highly promising for practical applications. Furthermore, the calculated reaction rate is 15.3 mol_C2H2

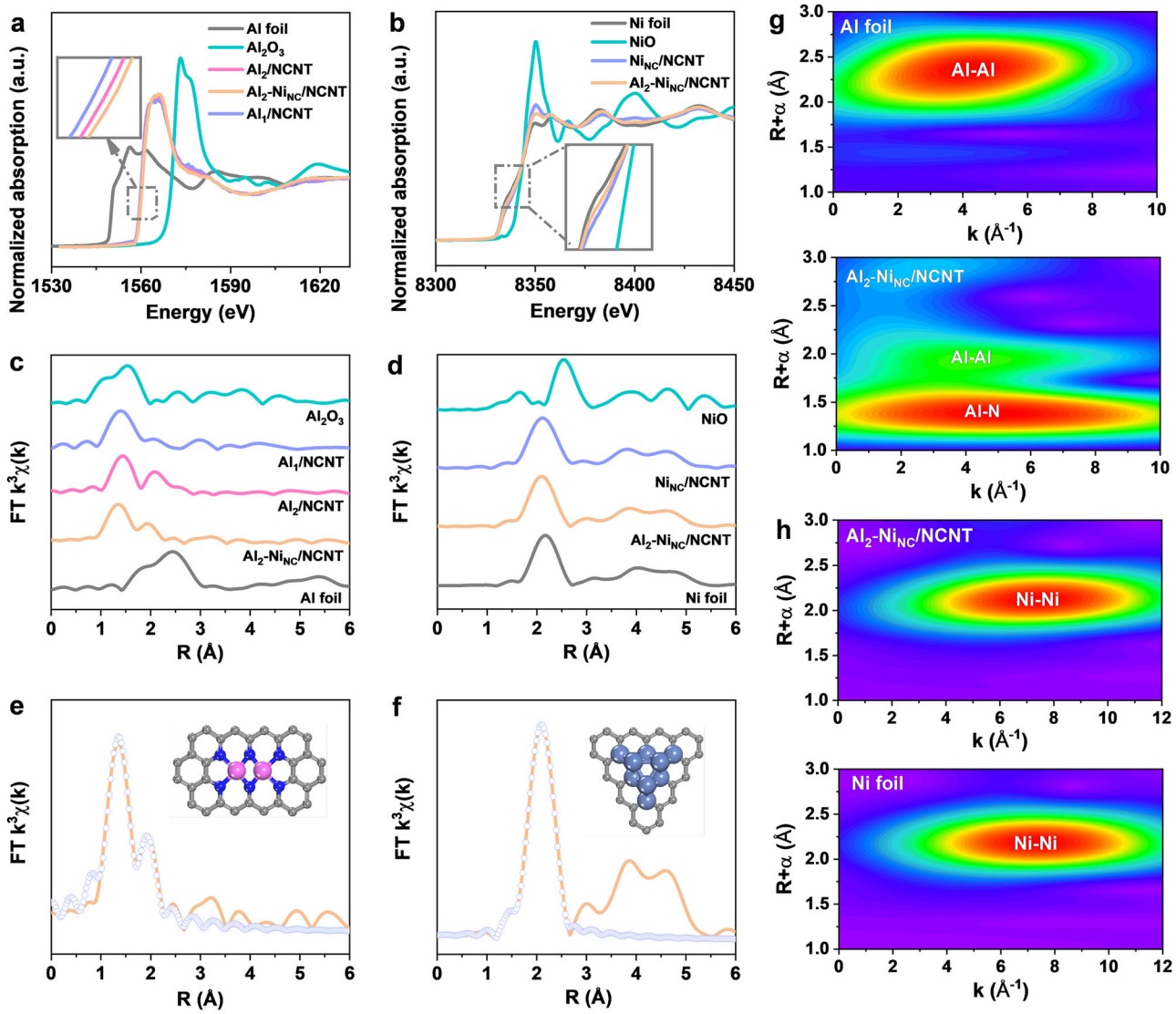

**Fig. 2 | Analysis of electronic structure and coordination environments. a** Al and (**b**) Ni K-edge XANES spectra; Fourier-transformed (**c**) Al and (**d**) Ni K-edge EXAFS spectra; (**e**) Al and (**f**) Ni K-edge EXAFS fitting in R space of Al$_2$-Ni$_{NC}$/NCNT (grey: C; blue: N; pink: Al; blue grey: Ni); (**g**) Al K-edge wavelet transform (WT)-EXAFS plots of Al foil and Al$_2$-Ni$_{NC}$/NCNT; (**h**) Ni K-edge wavelet transform (WT)-EXAFS plots of Ni foil and Al$_2$-Ni$_{NC}$/NCNT.

mol$_M^{-1}$ h$^{-1}$, approximately 2.4 times higher than Ni nanoclusters (Supplementary Fig. 20).

To clarify the kinetics advantages of the Al$_2$-Ni$_{NC}$/NCNT synergetic catalyst, the intrinsic activity, characterized by apparent activation energy (E$_a$) and reaction order was further analyzed under conditions below 15% conversion to eliminate the influence of mass or heat transfer. Firstly, kinetic measurements using the Arrhenius equation further affirm the intrinsic activity trend, with Al$_2$-Ni$_{NC}$ exhibiting a competitive advantage through a lower activation energy of 55.0 kJ mol$^{-1}$ (Fig. 3e). The reaction order (n) for acetylene was also determined by plotting intrinsic activity versus pressure (Fig. 3f), yielding a value of 0.19 for Al dual-atom sites integrated with Ni nanoclusters, while a negative value (−0.39) was observed for the aggregated Ni species. This negative reaction order suggests that high acetylene coverage inhibits the reaction rate due to strong adsorption[35]. Additionally, the hydrogen reaction order, shown in Fig. 3g, exhibits a linear relationship, with values of 0.39 for Al$_2$-Ni$_{NC}$/NCNT and 1.07 for Ni$_{NC}$/NCNT. The near-zero reaction order indicates that the catalytic activity of Al$_2$-Ni$_{NC}$ is determined by the structure rather than hydrogen pressure[36]. This implies that the synergy between

Al dual-atom sites and Ni nanoclusters effectively reduces competitive acetylene adsorption while enhancing hydrogen activation and dissociation.

Notably, Al$_2$-Ni$_{NC}$/NCNT demonstrates exceptional thermal and chemical stability under hydrogenation conditions, maintaining both activity and selectivity for at least 150 h at 142 °C, with no visible decline even upon regeneration (Fig. 3h). To investigate the nature and stability of this catalyst during reaction, XRD pattern and XAS analysis have been provided. As shown in Supplementary Figs. 21, 22, the atomically-dispersed Al sites and Ni clusters of Al$_2$-Ni$_{NC}$/NCNT could be maintained after the reaction; meanwhile, Al 2$p$ XPS results (Supplementary Fig. 23) further indicate that the average oxidation state of Al species well keeps between 0 and +3, following the coking combustion and a subsequent hydrogen activation, which is basically in agreement with the fresh one. Additionally, the excellent resistance to carbon deposition is also confirmed (Supplementary Fig. 24)[37]. In contrast, Ni$_{NC}$/NCNT shows a gradual decrease in ethylene selectivity, approximately with a loss of ~90% within the first five hours (Supplementary Fig. 25), and Raman measurements indicate severe coking (Supplementary Fig. 24)[37].

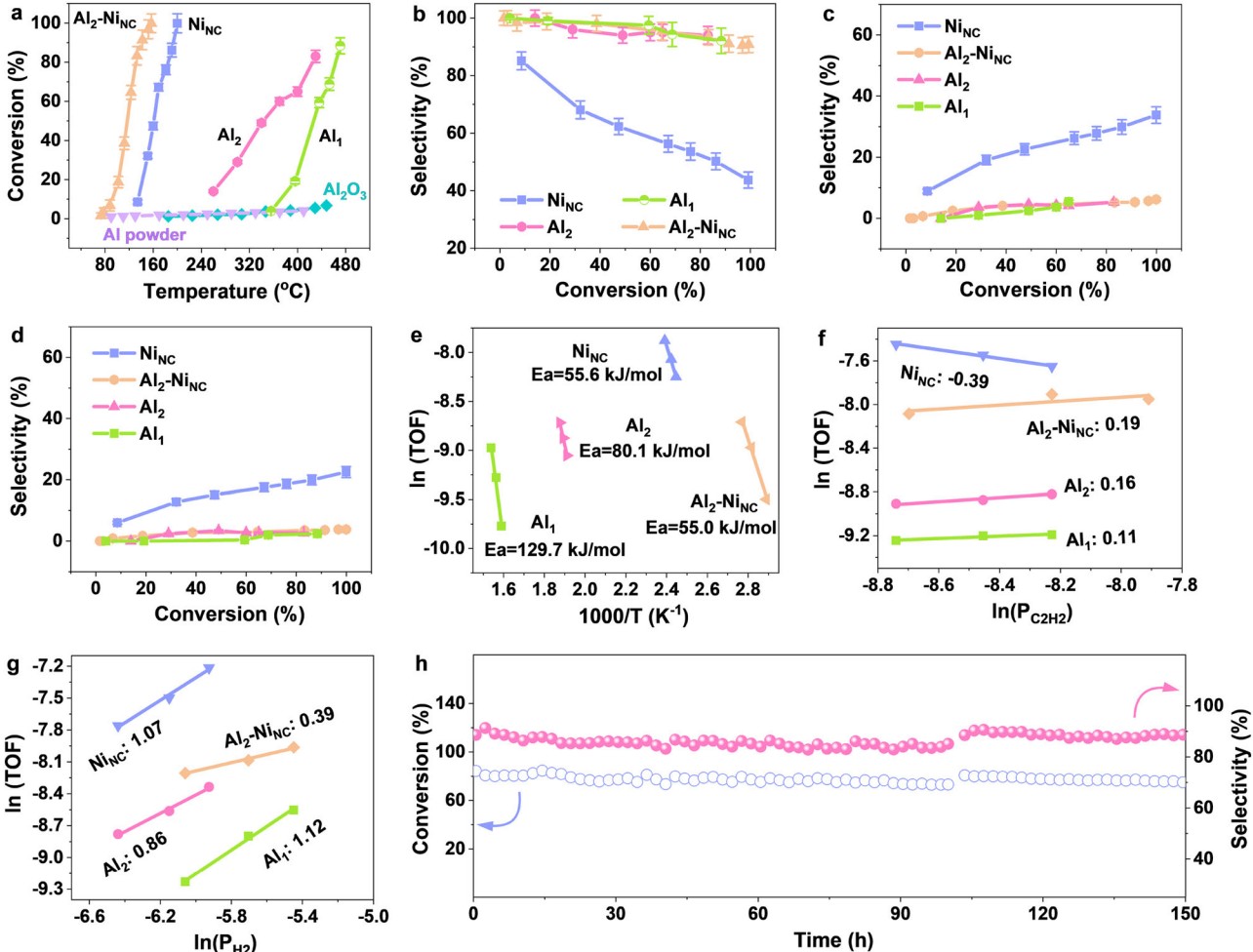

**Fig. 3 | Catalytic performance of acetylene hydrogenation. a** Conversion as a function of reaction temperature; Selectivity of (**b**) $C_2H_4$, (**c**) $C_2H_6$ and (**d**) oligomers as a function of conversion; (**e**) Arrhenius plots; (**f**, **g**) Reaction order of acetylene and hydrogen, respectively; (**h**) Durability test on $Al_2$-$Ni_{NC}$/NCNT at 142 °C. (Error bars represent the standard deviation of three repeating experiments; Reaction condition: a hydrogen to acetylene ratio of 20:1; space velocity = 9000 mL $g^{-1}$ $h^{-1}$; atmospheric pressure. The regeneration method originates from Jam domestic petrochemical's process for acetylene hydrogenation and primarily involves coke combustion through the introduction of oxygen at 500 °C, followed by a hydrogen treatment step to reduce the oxidized Al and Ni species back to their fresh states[44,45].

## Adsorption behavior and reaction mechanism

To investigate the role of different sites in $Al_2$-$Ni_{NC}$/NCNT catalysts, the adsorption and hydrogenation processes were monitored by in situ FT-IR spectroscopy. As shown in Fig. 4, acetylene is rapidly adsorbed onto the catalyst surface, reaching equilibrium with characteristic bands at 3309, 3262, 1353 and 1303 $cm^{-1}$. After purging with Ar, bands originating from gas-phase acetylene gradually disappear, while a new peak appears at 3280 $cm^{-1}$ for $Al_2$-$Ni_{NC}$/NCNT catalyst (Fig. 4a), indicating π-adsorbed acetylene[38]. When hydrogen is introduced, the intensity of π-binding reactant decreases significantly as reaction temperature rises (Fig. 4b), suggesting the efficient conversion of acetylene. Simultaneously, two characteristics at 1710 and 1606 $cm^{-1}$ appear, corresponding to the vibrations of C = C and =$CH_2$ in ethylene product in the π-bonded configuration[39,40]. The intensity of these peaks initially increases and then declines as the reaction temperature rises. More importantly, no bands originating from by-products, such as over-hydrogenated ethane (2940 and 2883 $cm^{-1}$) and polymerized oligomers (1794 and 1750 $cm^{-1}$), are detected in this process. This illustrates that the active sites, formed by Al dual-atom sites coupling with Ni nanoclusters in the $Al_2$-$Ni_{NC}$/NCNT catalyst, promote the selective conversion of acetylene to ethylene. This process exhibits excellent desorption capability, as confirmed by $C_2H_4$-TPD results

(Supplementary Fig. 26). Interestingly, in contrast to this behavior, acetylene is adsorbed on the Ni nanoclusters in dissociative and di-σ types (3250 and 1630 $cm^{-1}$, $Ni_{NC}$/NCNT in Fig. 4c), leading to the generation of by-products (Fig. 4d), consistent with the reported literature[41]. Comparing these two catalysts, it is evident that the introduction of Al dual-atom sites adjacent to Ni nanoclusters alters the adsorption configuration of reactants. Indeed, when using the Al dual-atom catalyst as a contrast, acetylene adsorbs in a π configuration on pristine Al dual-atom (Fig. 4e). This suggests that two adjacent Al species act as adsorption and activation sites for acetylene in $Al_2$-containing catalysts ($Al_2$-$Ni_{NC}$/NCNT and $Al_2$/NCNT), facilitating the formation of weakly adsorbed vinyl intermediates (-CH = $CH_2$). The hydrogenation of these intermediates is kinetically easier than C-C coupling[42] (Fig. 4b, f).

Though Al dual-atom promote ethylene formation, the adsorption peak of acetylene on the $Al_2$/NCNT catalyst decreases slowly after hydrogen is introduced, indicating low activity (50% @ 340 °C) due to the weak ability of Al species to activate $H_2$, which agrees with $H_2$-TPD results (Supplementary Fig. 27). However, despite having the same adsorption configuration of reactants, the $Al_2$-$Ni_{NC}$/NCNT drives the reaction faster (99.98% conversion @157 °C). This demonstrates that Ni nanoclusters more effectively dissociate $H_2$. In other words,

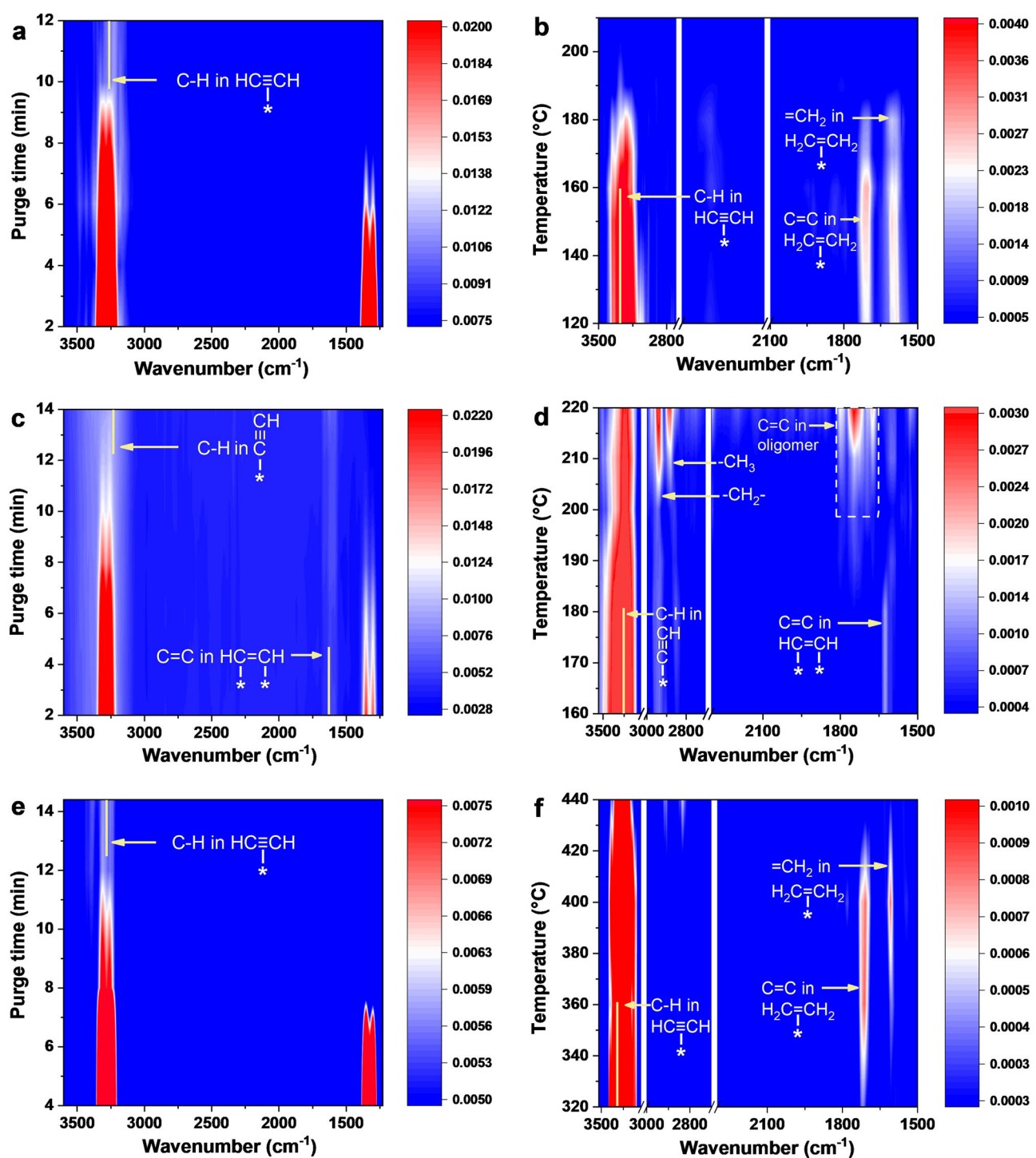

**Fig. 4 | Reaction mechanism of acetylene hydrogenation catalyzed by Al$_2$-Ni$_{NC}$/NCNT.** In situ FTIR spectra of the pre-adsorption and hydrogenation on (**a**, **b**) Al$_2$-Ni$_{NC}$/NCNT, (**c**, **d**) Ni$_{NC}$/NCNT and (**e**, **f**) Al$_2$/NCNT.

integrating Al$_2$ with Ni nanoclusters creates distinct roles: the Al dual-atom promote acetylene adsorption and activation in π binding, while Ni nanoclusters are responsible for the dissociation of H$_2$ into active H*, transferring to the generated -CH = CH$_2$ intermediates. This H transfer could be supported by a greater amount of H$_2$ consumption being 230 μmol/g than 189 μmol/g required from Ni nanoclusters (Supplementary Fig. 27). Meanwhile, an experiment was conducted in which WO$_3$ was mixed with Al$_2$-Ni$_{NC}$/NCNT, and a distinct color change from canary yellow to dark blue was observed (Supplementary Fig. 28), confirming the occurrence of hydrogen spillover[43]. These integrated sites alleviate the competitive adsorption of reactants at a single site, preventing over-hydrogenation and polymerization, thus overcoming the trade-off between activity and selectivity. As a result, and as expected, ethane and oligomers are prominently detected in the Ni$_{NC}$/NCNT catalyst, with no ethylene formation. This observation aligns well with temperature-programmed surface reaction (TPSR) results (Supplementary Fig. 29), which show that acetylene hydrogenation on Al$_2$-Ni$_{NC}$/NCNT starts at a lower temperature (77 °C). The process exclusively produces ethylene as the anticipated product, without generating ethane or oligomers, in contrast to Ni$_{NC}$.

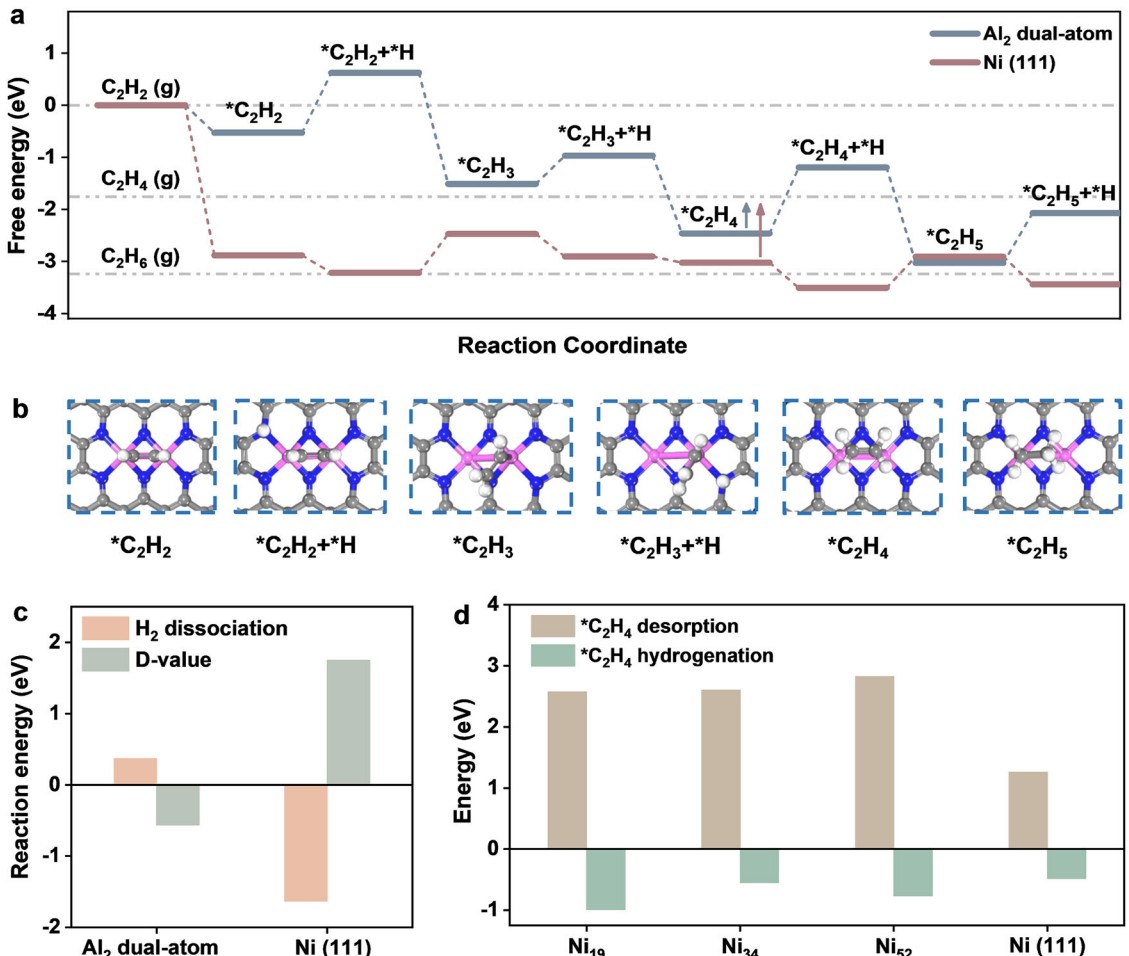

**Fig. 5 | Reaction mechanism of acetylene hydrogenation. a** Free energy profile of selective acetylene hydrogenation on Al₂ with Ni nanoclusters as contrast; (**b**) Configuration of reaction intermediates (grey: C; blue: N; pink: Al; white: H); (**c**) Energies of hydrogen dissociation and D-value standing for energy difference between C₂H₄ desorption and its further hydrogenation; (**d**) Reaction energy for ethylene desorption and hydrogenation on various Ni sites on Ni₁₉, Ni₃₄ and Ni₅₂ and Ni (111).

To further understand the high activity and selectivity towards ethylene of Al₂-Ni$_{NC}$/NCNT compared to Ni bulk, density functional theory (DFT) calculations were performed. Based on experimental structural characterizations, two optimal atomic structure models were constructed with the Al₂ and Ni nanoclusters (Figs. 2e and 2f). Several important pieces of information could be extracted in the free-energy profile of acetylene hydrogenation (Figs. 5a-5b, Supplementary Figs. 30 and 31). Primarily, C₂H₂ molecule coordinates on Al₂ sites in a π-complex manner with an adsorption energy of −0.53 eV. The resulting *C₂H₂ species is hydrogenated in successive steps to easily produce vinyl intermediate as the precursors of ethylene products. As expected, the generated *CH₂CH₂ intermediate adsorbs on Al₂ sites in a π-adsorbed configuration, which facilitates the desorption step with an energy of 0.70 eV. Indeed, this desorption energy is lower than that of further hydrogenation (referred to as D-value, which is negative in Fig. 5c), suggesting that acetylene hydrogenation favors ethylene rather than ethane. However, acetylene adsorption on Ni sites occurs in a multi-bridged configuration with a strong binding energy of −2.88 eV, which favors conversion to ethane (namely the D-value is positive in Fig. 5c). Notably, the differential charge density plots in Supplementary Fig. 32 clearly reveal the distinct adsorption behaviors of C₂H₂ and C₂H₄ on the Al₂ site and Ni (111) surface. On Al₂ site, the charge redistribution is predominantly localized between the C-C bond and the surface, indicating a weak interaction. In the case of C₂H₄, only slight surface polarization is observed, suggesting limited electronic coupling and facile product desorption. In contrast, adsorption of both C₂H₂ and C₂H₄ on Ni (111) surface induce pronounced charge transfer, wherein the C π electrons strongly interact with Ni d orbitals, resulting in significant charge redistribution and stronger binding between the adsorbate and the surface. Consequently, the weak adsorption on Al₂ facilitates the formation and release of C₂H₄, whereas the strong chemisorption on Ni (111) hinders C₂H₄ desorption and reduces the selectivity for C₂H₄ hydrogenation. Indeed, the Ni sites exhibit a significant increase in free energy during the *CH₂CH₂ desorption step compared to its hydrogenation (Fig. 5d and Supplementary Fig. 33) regardless of whether the systems involve clusters or bulk catalysts. Specifically, the desorption energy of *CH₂CH₂ on Ni sites of all sizes is markedly high, indicating strong adsorption. The subsequent hydrogenation of *CH₂CH₂ readily occurs on both Ni clusters of various sizes and Ni (111) surface, with reaction energies ranging from −0.99 to −0.48 eV. This implies that the multi-bridged adsorption configuration stabilizes the *CH₂CH₂ intermediate, rendering desorption energetically unfavorable and leading to poor ethylene selectivity. The above results are in good agreement with DRIFT analysis. Furthermore, the reaction energy for hydrogen dissociation has been calculated (Fig. 5c and Supplementary Fig. 34). Ni species display a stronger thermodynamic propensity for hydrogen activation than the Al₂ site, which is energetically less favorable, demonstrating that the Ni (111) surface is highly effective at activating H₂. Thus, Ni sites exhibit stronger adsorption for C₂H₂ and H₂

compared to Al sites. However, catalytic performance is governed not only by the adsorption of reactants but also by the free-energy change associated with the rate-determining step ($\Delta G_{RDS}$, namely activity) and the selectivity toward $C_2H_4$ and $C_2H_6$. As calculated (Supplementary Fig. 35a), the $\Delta G_{RDS}$ value on Al sites is lower than Ni sites. Furthermore, $*C_2H_4$ hydrogenation is thermodynamically less favorable compared to $*C_2H_4$ desorption on Al sites, whereas it is more facile on Ni sites (Supplementary Fig. 35b). These thermodynamic analyses indicate that Al sites facilitate more favorable catalytic kinetics for the selective hydrogenation of acetylene, which is further supported by our calculated turnover frequency (TOF) values (Supplementary Fig. 36). Therefore, despite weaker adsorption of reactants, the superior intrinsic activity and enhanced selectivity of Al sites suggest that they serve as the primary active centers for acetylene selective hydrogenation.

To gain deeper insight into the adsorption behavior at Ni sites, the adsorption energies of $*C_2H_2$ and $*H$ under various $*C_2H_2$ coverages were calculated (Supplementary Figs. 37–39). As the coverage increases from 0.125 to 0.875 monolayer (ML), the adsorption energy of $*C_2H_2$ shifts from initial −3 eV to +0.2 eV, indicating progressive surface saturation with adsorbed $*C_2H_2$ species. In contrast, although $*H$ adsorption energy exhibits a slight increase with increasing coverage, it remains negative, implying that $*H$ species can still be generated from $H_2$ dissociation. Therefore, we propose that while Ni sites exhibited strong adsorption towards $*C_2H_2$ and $*H$, the favorable $*C_2H_2$ adsorption mainly leads to high coverage. In contrast, the readily formed $*H$ species could migrate to adjacent Al sites, where they participate in the hydrogenation of $*C_2H_2$. Based on the results from in situ spectroscopic characterizations and theoretical calculations, it could be regarded that the construction of the synergistic Ni nanoclusters and $Al_2$ active centers offers excellent activity and selectivity, in which nickel species are responsible for hydrogen dissociation, while the $Al_2$ sites trigger the spillover of active $H*$ to react with π-bound acetylene and weaken the desorption of ethylene instead of simply modify Ni, inhibiting over-hydrogenation and polymerization.

## Discussion

In summary, we explored a strategy to overcome the trade-off effect in the selective hydrogenation of acetylene by leveraging the synergy between Al dual-atom sites and Ni nanoclusters. Firstly, N-doped porous carbon nanotube was utilized to capture volatile $Al_2Cl_6$ molecules generated by the vaporization of anhydrous $AlCl_3$ (s). These $Al_2Cl_6$ molecules were subsequently converted to $Al_2$/NCNT under hydrogen annealing. Following the deposition of $NiCl_2$ vapor, we successfully synthesized the $Al_2$-$Ni_{NC}$/NCNT catalyst, which incorporates atomically dispersed $Al_2$ sites and Ni nanoclusters, as confirmed by various spectroscopic and microscopic characterizations. This $Al_2$-$Ni_{NC}$/NCNT catalyst exhibits excellent catalytic performance, enabling the selective conversion of acetylene to ethylene even under front-end ($H_2$-rich) reaction conditions, with no visible activity decay after 100 h continuous testing. Both experimental evidence and theoretical calculations demonstrated that this synergistic catalyst facilitates acetylene hydrogenation via vinyl intermediates to ethylene, without promoting polymerization or ethylene hydrogenation. This study not only firstly reports that atomically dispersed Al species could be used to catalyze the acetylene hydrogenation, but also creates a novel design strategy for a highly efficient catalytic system that integrates well-defined dual-atom structures with metal nanoclusters to enhance the catalytic performance.

## Methods
### Materials
All chemicals including nitrogen-doped carbon nanotube (NCNT, Macklin, > 95%), anhydrous aluminum chloride ($AlCl_3$, Innochem, 99.99%), alumina ($Al_2O_3$, Macklin, 99.99%), aluminum powder

(Aladdin, > 99.5%) and anhydrous nickel chloride ($NiCl_2$, Innochem, 99%) were of reagent-grade quality. They were acquired from commercial suppliers and employed as received. Deionized water with a specific resistance of 18.25 MΩ·cm was used.

### Preparation of $Al_2$-$Ni_{NC}$/NCNT catalyst with $Al_2$ dual-atoms and Ni clusters as comparison

$AlCl_3$ (150 mg) and NCNT (50 mg) were placed in two separate porcelain boats, the former of which was in the upstream. The powders were heated up to 180 °C in a slow-flowing Ar stream (20 mL min$^{-1}$) for 2 h to achieve the transfer and capture of $Al_2Cl_6$ vapor, followed by 5% $H_2$/Ar treatment at 550 °C for 2 h to obtain dual-atom Al catalyst ($Al_2$/NCNT). Subsequently, a second vapor deposition step was further conducted, in which anhydrous $NiCl_2$ (50 mg) and $Al_2$/NCNT (100 mg) were placed in a tubular furnace, with the former at the upstream position, followed by heating to 950 °C and maintaining for 1 h in Ar to obtain the final product, namely $Al_2$-$Ni_{NC}$/NCNT. As comparison, $Ni_{NC}$ catalyst was acquired using the above preparation method but with 50 mg of anhydrous $NiCl_2$ and 100 mg of NCNT.

### Preparation of $Al_1$ single-atom catalyst

NCNT (100 mg) were dispersed in 16 mL of deionized water via ultra-sonication for 30 min to obtain a homogeneous suspension. Under vigorous stirring, 4 mL of an aqueous solution containing $AlCl_3$ (15 mg) was slowly added dropwise to the NCNT suspension over 30 min. The mixture was then stirred at room temperature for 12 h. Following the reaction, the solid product was collected by centrifugation and dried for 12 h. The resulting powder was subsequently treated at 550 °C for 2 h under a flow of 5% $H_2$/Ar, yielding the $Al_1$/NCNT.

### Characterizations

Morphologies of as-prepared samples were investigated using transmission electron microscopy (TEM, FEI Tecnai G2 20 STwin, operated at 200 kV), and high-resolution transmission electron microscopy (HRTEM, JEOL 2100, operated at 200 kV). Aberration-corrected high-angle annular darkfield scanning transmission electron microscope (HAADF-STEM) results were obtained on a JEOL JEM-ARM200F STEM with a spherical aberration corrector (operated at 200 kV). The loadings of Ni and Al were conveyed by the inductively coupled plasma optical emission spectroscopy (ICP-OES) on Shimadzu ICPE-9800. The crystalline structures of all catalysts were evaluated using Shimadzu XRD-6000 diffractometer (Cu Kα source, $\lambda = 1.5418$ Å). The $N_2$ isothermal adsorption/desorption was performed on a Micromeritics ASAP 2460 system. Prior to $N_2$ adsorption, the powders were degassed at 200 °C for 3 h under vacuum. Raman spectra were identified on an inVia Reflex RamanMicroscope. Thermogravimetric mass spectrometry (TG-MS) was conducted with TGA-8000 thermogravimetry and Clarus SQ8T mass spectrometer. $H_2$ or $C_2H_4$ temperature-programmed desorption measurements were conducted on Micrometrics Autochem II 2920 chemisorption system equipped with a thermal conductivity detector (TCD). About 100 mg of the sample was purged with pure Ar or He at 200 °C and then cooled down to room temperature. $H_2$ or $C_2H_4$ at a flow rate of 50 mL/min was introduced to the sample for 1 h to ensure a saturated adsorption, and then pure Ar or He was flowed to remove gas phase. The TPD profiles were subsequently obtained by increasing the temperature from 50 to 800 °C at a rate of 10 °C/min. Temperature-programmed surface reaction (TPSR) experiment was performed in a reaction cell. 0.2 g of sample was pretreated in $N_2$ for 30 min. Then, the gas was switched to 0.31% $C_2H_2$/ 30.40% $C_2H_4$/6.20% $H_2$ (balanced with nitrogen) with flow rate of 30 mL/min and heated from 40 to 380 °C for data recording. The products (m/z of $C_2H_2$, $C_2H_4$ and $C_2H_6$ being 26, 27, and 15, respectively) were measured via mass spectrometer. Diffuse reflectance infrared Fourier transform spectra (DRIFTS) were collected on a Bruker TENSOR II with an in-situ diffuse reflection accessory. Typically,

about 50 mg of powders was purged with pure $N_2$ in a reflection cell at 200 °C and then cooled down to room temperature. Thereafter, 5.0 vol% $C_2H_2$/$N_2$ was introduced into the cell with a flow rate of 20 mL/min for 1 h to ensure the saturated adsorption of acetylene on the sample. Afterwards, pure $N_2$ was flowed to remove gas phase to obtain pre-adsorbed acetylene. Then, 10 vol% $H_2$/$N_2$ with a flow rate of 20 mL/min was flowed to the cell for hydrogenation of the pre-adsorbed acetylene, during which the temperature was increased gradually to 220 °C (for $Al_2$-$Ni_{NC}$/NCNT and $Ni_{NC}$ catalysts) or 440 °C (for $Al_2$/NCNT). The spectra collected at different temperature were subtracted from background spectra at the corresponding temperature. X-ray absorption spectroscopy (XAS) measurement and data analysis: The Ni and Al K-edge XAS experiments were performed on beamline 1W1B at the Beijing Synchrotron Radiation Facility (BSRF, Beijing, China). The Ni and Al K-edge data of different catalysts were recorded in a fluorescence mode, and the references (NiO and Ni foil; $Al_2O_3$ and Al foil) were recorded in a transmission mode. Data reduction, data analysis, and EXAFS fitting were performed with the Athena, Artemis, and IFEFFIT software packages.

## Catalytic testing

Catalytic behavior was measured using a fixed-bed microreactor at 0.1 MPa and 40–440 °C. 0.2 g of catalyst was diluted with quartz sand (Aldrich, 40-70 mesh) and placed in the reactor of 8 mm. The introduced feed gas consisted of a 0.31% $C_2H_2$/30.40% $C_2H_4$/6.20% $H_2$/1% $C_3H_8$ mixture diluted by nitrogen with a gas hourly space velocity (GHSV) of 9000 mL $h^{-1}$ $g^{-1}$. The reactant and product concentrations were analyzed by online Gas Chromatography (GC) with a flame ionization detector using a PLOT capillary column (50 m × 0.53 mm). Propane was used as an internal standard. Multiple data points were collected at different temperatures to ensure reproducibility. The selectivity to oligomers was calculated based on carbon balance (100 ± 0.5%). Acetylene conversion and ethylene selectivity were calculated as follows:

$$\text{Acetylene conversion(\%)} = \frac{C_2H_2(\text{inlet}) - C_2H_2(\text{outlet})}{C_2H_2(\text{inlet})} \quad (1)$$

$$\text{Ethylene selectivity(\%)} = \frac{C_2H_4(\text{outlet}) - C_2H_4(\text{inlet})}{C_2H_2(\text{inlet}) - C_2H_2(\text{outlet})} \quad (2)$$

In which $C_2H_2$ (inlet), $C_2H_4$ (inlet), $C_2H_2$ (outlet) and $C_2H_4$ (outlet) represent the concentration of reactant and product. Catalyst stability was performed for a long period at a constant temperature of 142 or 181 °C.

## Regeneration

The used $Al_2$-$Ni_{NC}$/NCNT catalyst by oxidative treatment to remove oligomers, following a protocol adapted from industrial practices used at Jam domestic petrochemical for acetylene hydrogenation[44,45]. Firstly, a nitrogen stream at a flow rate of 100 mL $min^{-1}$ was fed to the reactor at 115 °C for 60 min. The temperature was then increased to 165 °C while maintaining the nitrogen flow to stabilize the catalyst bed. In the next step, in addition to nitrogen, water vapor ($H_2O$ (g)) at a flow rate of 0.02 mL $min^{-1}$ was injected as the temperature was raised from 165 to 400 °C, and this temperature was maintained for 90 min to wash out and reform light hydrocarbons. Subsequently, the heavy hydrocarbons prone to coking were burnt by introducing oxygen with the flow rate gradually increased from 5 to 40 mL $min^{-1}$ at 500 °C. Following this step, nitrogen was purged through the reactor for 130 min to displace residual oxygen. Then, a subsequent hydrogen stream at 10 mL $min^{-1}$ was introduced to reduce the oxidized Al and Ni species back to their fresh states. Finally, nitrogen was passed through the system until the bed temperature decreased to ambient conditions, yielding the regenerated $Al_2$-$Ni_{NC}$/NCNT catalyst.

## Computational details

Density Functional Theory (DFT) calculations were conducted using Vienna Ab initio Simulation Package (VASP)[46], the version number of which is 5.4.1. Interactions between the ionic core and valence electrons were modeled complementing the projector-augmented wave (PAW) method[47,48]. The Perdew-Burke-Ernzerhof (PBE) exchange-correlation within the framework of the generalized gradient approximation (GGA) functional was employed[49]. The convergence criteria for force and energy were set to 0.02 eV/Å and $10^{-6}$ eV, respectively. The Brillouin zone was sampled in $k$-space using a Monkhorst-Pack $k$-point grid mesh of $3 \times 3 \times 1$ for geometric optimization. A 15 Å vacuum layer was used to reduce virtual interactions between slabs along the z direction. Considering the presence of a magnetic element (Ni) in the system, the spin-polarized effect was incorporated. The work of Kresse and Hafner has indicated that surface magnetism is vital for an accurate quantitative analysis of adsorption energy[50].

The carbon-carbon bond length within graphene was optimized to 1.42 Å and the in-plane lattice constant of graphene to $2.64 \times 2.64$ Å$^2$, while the Ni crystal lattice constant was refined to 3.52 Å, both values closely matching with established experimental values[51,52]. For Ni surface, a p ($4 \times 4$) four-layer supercell was used, while fixing the bottom two layers and relaxing the upper two layers. While a p ($6 \times 6$) monolayer graphene was utilized to build the $Al_2$, and a p ($8 \times 8$) graphene supercell was used to support the $Ni_{19}$, $Ni_{34}$, and $Ni_{52}$ clusters. Elemental steps for acetylene hydrogenation were considered as listed below:

$$* + C_2H_2(g) \rightarrow {}^*C_2H_2 \quad (3)$$

$$* + \frac{1}{2}H_2(g) + {}^*C_2H_2 \rightarrow {}^*C_2H_2 + {}^*H \quad (4)$$

$${}^*C_2H_2 + {}^*H \rightarrow {}^*C_2H_3 \quad (5)$$

$$\frac{1}{2}H_2(g) + {}^*C_2H_3 \rightarrow {}^*C_2H_3 + {}^*H \quad (6)$$

$${}^*C_2H_3 + {}^*H \rightarrow {}^*C_2H_4 \quad (7)$$

$$\frac{1}{2}H_2(g) + {}^*C_2H_4 \rightarrow {}^*C_2H_4 + {}^*H \quad (8)$$

$${}^*C_2H_4 + {}^*H \rightarrow {}^*C_2H_5 \quad (9)$$

$$\frac{1}{2}H_2(g) + {}^*C_2H_5 \rightarrow {}^*C_2H_5 + {}^*H \quad (10)$$

$${}^*C_2H_5 + {}^*H \rightarrow {}^*C_2H_6 \quad (11)$$

$${}^*C_2H_6 \rightarrow * + C_2H_6 \quad (12)$$

Where * represents a surface site, the reaction energy along elemental steps of acetylene hydrogenation is calculated to investigate the reaction process, especially the selectivity, that is proceeding the ${}^*C_2H_4$ desorption for $C_2H_4$ production and subsequent hydrogenation for $C_2H_6$.

Besides, adsorption energies are calculated as below, which is used to evaluate the adsorption ability:

$$\triangle E_{ads} = E_{ads/surf} - E_{surf} - E_{ads} \quad (13)$$

Where $\triangle E_{ads}$ is the adsorption energy, $E_{ads/surf}$ is the total energy of the adsorbates adsorbed on the surface, $E_{surf}$ and $E_{ads}$ are the energies of the isolated slab and molecule, respectively.

The free energy of all species involved in acetylene selective hydrogenation is calculated below:

$$G = E + E_{zpe} + PV - TS \qquad (14)$$

Where G represents Gibbs free energy, while E and $E_{zpe}$ are the electronic energy via DFT calculations and zero-point energies, respectively. The pressure-volume (PV) contribution is considered negligible[53]. The temperature (T) is considered as 298 K, and S is the entropy.

The dissociation energy of $H_2$ is calculated in the following equation:

$$E_{dis} = E_{*2H} - E_* - E_{H_2} \qquad (15)$$

$E_{dis}$ represents the dissociation energy, where $E_{*2H}$ is the total energy of the catalyst after adsorbing two H atoms, $E_*$ is the energy of the clean catalyst surface, and $E_{H_2}$ is the energy of the gas-phase $H_2$ molecule. This energy reflects the difficulty of $H_2$ dissociation on the catalyst surface. a more negative value indicates that $H_2$ molecules dissociate more easily on the surface.

The desorption energy of $C_2H_4$ ($E_{des}$) is calculated below:

$$E_{des} = E_* + E_{C_2H_4} - E_{*C_2H_4} \qquad (16)$$

Here, $E_{C_2H_4}$ represents the energy of the $C_2H_4$ molecule in the gas phase, and $E_{*C_2H_4}$ denotes the total energy of the catalyst surface with an adsorbed $C_2H_4$ molecule. $E_{desorption}$ represents the energy required for the desorption of $C_2H_4$ from the catalyst surface; a more negative value of $E_{desorption}$ indicates that $C_2H_4$ can desorb more easily from the surface.

The hydrogenation energy of $C_2H_4$ ($E_{hyd}$) is calculated below:

$$E_{hyd} = E_{*C_2H_5} - E_{*C_2H_4} - \frac{1}{2}E_{H_2} \qquad (17)$$

Where $E_{*C_2H_5}$ represents the total energy of the catalyst surface with an adsorbed $C_2H_5$ species, and $E_{hydrogenation}$ denotes the energy required for the hydrogenation of $C_2H_4$ to form $C_2H_5$. A more negative $E_{hydrogenation}$ value indicates that the hydrogenation of $C_2H_4$ to $C_2H_5$ is more favourable.

The definition of the D-value is as follow:

$$D \text{ value} = E_{desorption} - E_{hydrogenation} \qquad (18)$$

The more negative this value is, the easier the catalyst is to remove the $C_2H_4$ molecules, and the more difficult hydrogenation occurs.

## Data availability

Data will be made available from corresponding author on request.

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

## Acknowledgements

This work was financially supported by the National Natural Science Foundation of China (22278017 [Y.L.], 22475017 [Y.Z.], 22471012 [S.T.]), Innovation Fund of SINOPEC Catalyst Co. Ltd-State Key Laboratory of Chemical Resource Engineering (No. 36100000-22-ZC0607-0041 [Y.L.]), Fundamental Research Funds for the Central Universities (JD2508 [Y.L.]), and the Young Elite Scientists Sponsorship Program by BAST (No. BYESS2023087 [Y.L.]). The computational study is supported by the Marsden Fund Council from Government funding (21-UOA-237 [Z.W.]) and Catalyst: Seeding General Grant (24-UOA-048-CSG [Z.W.]), managed by Royal Society Te Apārangi. All DFT calculations were carried out on the New Zealand eScience Infrastructure (NeSI) high-performance computing facilities. M. L. thanks the program of China Scholarships Council for financial support.

## Author contributions

Y.L. and H.Y. carried out most of the characterization, structural analysis, and catalytic reactions. M.L. and R.L. carried out DFT calculation. L.Y. prepared the catalysts and performed some characterizations of catalysts. X.F., Z.Z. and Y.Z. helped to analyze the data. Y.L., Z.W. and S.T. designed the experiment, discussed the results, wrote the manuscript and provided the funding support.

## Competing interests

The authors declare no competing interests.
