## [Transparent Peer Review file · Nature Communications]

Synergistic Aluminum Dual-Atom Sites and Nickel Nanoclusters for Acetylene Selective hydrogenation

Corresponding Author: Professor Shubo Tian

Version 0:

Reviewer comments:

Reviewer #1

(Remarks to the Author)

Liu et al. reported a synergistic catalytic strategy by integrating two-functional Al dual-atom sites (Al₂) and Ni nanoclusters (NiNC) to enable highly selective hydrogenation of acetylene. In this system, as authors reported, the Al sites weakly adsorb acetylene and catalyze the subsequent hydrogenation process, while the Ni sites facilitate hydrogen spillover for its subsequent hydrogenation. The resulting Al₂-NiNC/NCNT catalyst demonstrated excellent selectivity toward ethylene, maintaining approximately 90% selectivity and exhibiting outstanding long-term stability without noticeable activity loss after 100 hours of continuous operation. Compared to strategies in previous studies focusing on the modifications in single-atom catalysts (SACs), nanoparticle systems, or support engineering (Chem. Rev. 2020, 120, 683-733; JACS 2023, 145, 26728-26735), this work presents a novel intergrade strategy through combining Al₂ and NiNC in carbon supports, significantly suppress the over-hydrogenation to ethane. Notably, the strategy is compelling due to its effective integration of distinct active sites, rather than relying on precise structural or electronic modulation. Based on this, I suggested this work can be considered for acceptance, but I still have several questions on this work for further revision as follows.

1. In this work, I primarily interested in why the acetylene can adsorbed on Al sites for subsequent hydrogenation, rather than Ni sites. As we know, the Ni sites can enable strong adsorption compared to Al sites, which would appear to be more energetically favorable for acetylene adsorption.

2. Figure 5 presents DFT calculated results, including reaction energies, adsorption energies, and notably the D value, which is important for elucidating the underlying mechanism. Given that these results serve different roles in illustrating the reaction mechanism, detailed definitions and corresponding descriptions should be provided in the main text to aid reader comprehension.

3. Figure 5 provides thermodynamic results to explore the role of Ni and Al sites in hydrogenation of acetylene, but it is not clear to describe particularly given the energy values dependent on DFT methodologies. I would recommend including additional electronic structure analyses, which could provide more direct insight into the interaction between acetylene and the active sites.

4. The atom labeling in the visualizations of models appears to be missing or insufficient. Clear labeling of key atoms is crucial for readers.

5. As author calculated, the behavior on various scale of Ni models were studied to understand the role of Ni sites. But the size of catalysts can affect the catalysis, which is commonly reported (J. Catal. 2023, 425, 70–79). Therefore, the discussion on particle size effect should be considered, whether the size effect is not affected the results, especially the H₂ dissociation and ethylene adsorption.

6. Compared to Al₂O₃, Al₂/NCNT and Al₁/NCNT have significantly improved activity. It is recommended to supplement the performance experiment of NCNT to rule out the interference of the support on atomically precise Al catalysts.

7. The conversion should be controlled below 15% for Arrhenius plot measurement. This is critical to validate the reliability of the activation energy and reaction order. The authors should clarify this point. Meanwhile, the authors claim that atomically precise Al catalysts exhibit good activity for the acetylene hydrogenation, but missing the sufficient kinetic analysis, especially for Al₁ catalysts, which is important for catalytic mechanism.

Reviewer #2

(Remarks to the Author)

The authors of this manuscript propose an intriguing concept, where atomically dispersed Al sites (in pairs) work in concert with Ni nanoclusters to achieve high selectivity for acetylene hydrogenation, an industrially relevant reaction. The use of atomically dispersed Al on carbon nanotubes is novel, however I would like to know if the catalyst will survive any oxidation which may be required to burn off the oligomers in an industrial setting.

Also, I have a few concerns about the manuscript which should be addressed by the authors:

1) Metallic Ni is not a selective catalyst for this reaction and leads to over hydrogenation. So, if you have any metallic clusters exposed, they will lower the selectivity of the reaction. How do you keep the Ni from doing the hydrogenation (non-selectively) since it is a more active catalyst than the Al. From the schematic diagram of the synthesis, in Figure 1, it appears that the Ni is added after the addition of the Al. So, the Ni is exposed to the gas phase and can potentially catalyze the non-selective over hydrogenation to ethane.

2) The AC-STEM images need revision.

a. First, Figure 1c is too dark. It needs to be brightened to be able to see clearly. Also, a corresponding image without the circles needs to be included side by side with Figure 1c, so the reader can see clearly the basis for choosing to circle the atoms that are being highlighted.

b. Next, the authors circle some bright objects on the surface of the Ni nanoparticles in Figure 2d. As in comment 2a above, the authors need to show in the supporting information this image along with one without the circles. And they need to explain the basis for concluding there are Al atoms on the surface of the Ni. Al is much lighter than Ni, furthermore similar bright objects are also seen on the Ni catalyst (Figure 1e). So, without some other evidence, perhaps XPS, or LEIS and also EDS, we cannot be sure that the Ni is indeed covered by the Al atoms.

3) The analysis of the EXAFS via fitting the data does not show any evidence for Ni nearest neighbors to the Al, or vice versa. But this is implied in Figure 1d. Do the Ni clusters expose metallic Ni atoms to the gas phase? If so, what stops them from doing the non-selective hydrogenation of ethylene to ethane. Are the Ni clusters physically distant from the Al atoms on the carbon.

4) There are several aspects of the catalytic measurements that need clarification:

a. It is stated that the Al₂/NCNT catalyst shows no influence of temperature on the conversion or selectivity. Does the reactor system have any background reactivity? It seems odd that there is increase in rate with temperature.

b. In the supporting information, they need to show more details of the reaction products, for instance the concentrations of acetylene, ethylene and ethane as a function of temperature, or a function of space velocity, so the reader can see when the formation of ethane starts and what happens as acetylene is fully consumed. Figure 3 only shows conversion and selectivity. It is important to see all the products and to understand how much ethane is being produced.

c. Did they quantify the rate of formation of oligomers, or green oil during their long term tests.

Reviewer #3

(Remarks to the Author)

This manuscript reports the synthesis of Al dual-atom sites and Ni clusters (ca. 2.5 nm average size) on nitrogen-doped carbon nanotubes (Al₂-NiNC-NCNT) and its catalytic activity for the selective semi-hydrogenation reaction of acetylene in ethylene streams. The corresponding blanks (Al₂-NCNT, NiNC-NCNT and also a Al-NCNT solid catalyst) are also prepared and tested. The reactivity shows that while Al₂-NiNC-NCNT and NiNC-NCNT are not far in catalytic activity (confirmed by the similar activation energy), the former is more selectivity, since apparently Al avoids over-hydrogenation of acetylene to ethane (and coupling reactions) on the Ni cluster sites. The latter is due to a much higher adsorption of acetylene on the Ni sites than in Al, favouring the undesired further reactions. If this is so, it means that Al overrules Ni when activating acetylene, and this is something that is not clear in the study. It could be that H₂ competes strongly with acetylene on Ni sites but, in this case, the catalytic activity of NiNC-NCNT would be much lower, and it is not, just the selectivity. Another point that makes the key claim of the study, i.e. that the symbiosis between dual Al sites (perform all the process except H₂ dissociation) and Ni (H₂ dissociation) triggers the reaction, to be doubtful is the absence of any significant characterization of the catalyst during or after reaction. The only data given are just a Raman, where any information on the metal sites cannot be obtained. The atomically-dispersed Al sites could be rapidly destroyed (aggregated, sublimated) under the thermal reaction conditions, even the Ni clusters could further aggregate or disperse. If the main advance of the paper is the originality of the catalytic sites and the mechanism, the nature and stability of the fresh catalyst during reaction must be clearly demonstrated.

With that said, other issues could be considered less important, but nor for the high standards of Nat. Commun.:

1) Catalytic results: The conversion of acetylene is given in rough numbers (100%), that means 99%, 99.99%? The industrial process requires less than 1 ppm in the final stream, although <10 ppm can be acceptable at the academic level. This has to be measured and explain how is measured (exact mass balance, not with a 0.5% error). A 10% of ethane is very much, the industrial process requires less than 2%, otherwise the economic penalty is huge.

2) Catalyst nature: Why using CNTs if the support structure does not have any influence on the formation of the metal sites, as demonstrated by the Raman experiments? In this case, a simple N-doped carbon should be enough. Is N doping really essential? One could argue that O instead of N is stabilizing the dual sites (even perhaps Cl). The difference if XAS between

N and O is very low. N XPS should help here. Any evidence that all the Cl is gone? It could still be on the catalyst.

The Al1-catalyst is made by impregnation in water solution, and no characterization is given. The likely structure here is Al₂O₃ entities, thus comparison with the Al2-catalyst is quite doubtful.

In any case, any Al-N or Al-O single atom site is quite prone to decompose under H₂ atmospheres and severe heating, thus shedding more doubts on the catalyst stability under reaction conditions.

Figure 1b (PXRD) does not guarantee the formation of Ni aggregates, it looks the opposite.

3) Reaction conditions: The reaction temperatures employed here (>150 °C, >300 °C for only Al) are extremely high for this reaction. The optimized reaction conditions employ 20:1 H₂:ethylene, this is much more than standard front-end conditions.

Non-precious metal catalysts can perform the reactions under milder reaction conditions, see for instance J. Am. Chem. Soc. 2018, 140, 8827–8832 (this article should be cited and included in the comparative Table). This comparative Table is partially informative, it is true that hundreds of catalytic systems have been published, but the selection here is quite random, and the commercial catalyst (Pd-Al₂O₃) operates at much milder reaction conditions (35-70 °C) than indicated in the Table.

4) Reaction mechanism: The characterization (HR-TEM) claims that the dual Al sites are nearby the Ni clusters, surrounding them. There is not clear evidence of this (EDS mapping, Al-Ni reduced distances in XAFS, the XPS shift is very doubtful). In any case, H-spillover from Ni to Al sites would not require any spatial approximation, and that has not been considered. Please check (a simple test with WO₃ will confirm the H spillover).

Minor comment: The meaning of TPSR should be specified (not said neither in the main text or SI).

In conclusion, although recognizing that the activation of acetylene on atomically-dispersed Al sites for the semi-hydrogenation reaction is of academic interest, the study here does not demonstrate that this is occurring, since the stability of the Al site during reaction and a significant catalytic performance beyond that of the Ni sites, is not demonstrated.

Version 1:

Reviewer comments:

Reviewer #1

(Remarks to the Author)

I have carefully reviewed the revised manuscript and the authors' detailed responses to my comments. They have made substantial and satisfactory efforts in addressing all the raised issues. The manuscript has been significantly improved and is now clear, compelling, and ready for publication.

Therefore, I am pleased to recommend its acceptance in its current form.

Reviewer #2

(Remarks to the Author)

The authors have revised the manuscript and have addressed my concerns adequately.

Reviewer #3

(Remarks to the Author)

I recognize that the authors have made an effort to answer to all Reviewer's requirements, including mine. However, two main issues still puzzle me. Regarding characterization of the spent catalyst, the only technique shown here that gives real information about the isolated Al atoms is the XAS, but the spectra are presented in a very general way, without comparing with the fresh one. This has to be done. Second, and more important, one of the Reviewers insightfully asked about the regeneration of the solid catalyst with O₂, to burn the residual carbonaceous molecules. The authors claim that the isolated Al sites are able to bear the treatment with O₂ at 500 °C without oxidation. This is completely unbelievable, if so, this is the discovery here: A solid that keeps Al unoxidized at 500 ° in O₂. Catalysis with O₂ should be tested! Trying to be positive here, the only explanation that comes to my mind in order to accept that the catalyst preserves structure and activity after treatment with O₂ at 500 °C, is that the Al sites are originally oxidized and that, under the H₂ conditions and with the Ni sites continuously supplying H atoms by spillover during reaction, the Al sites become active for acetylene. However, the most plausible explanation is that Al has nothing to do with acetylene activation, but exerts some kind of modification on Ni. All these arguments should be considered by the authors before publication.

Version 2:

Reviewer comments:

Reviewer #3

(Remarks to the Author)

The questions have been properly answered, the manuscript seems ready for publication.

Response to reviewers

Reviewer #1

Liu et al. reported a synergistic catalytic strategy by integrating two-functional Al dual-atom sites (Al_2) and Ni nanoclusters (Ni_{INC}) to enable highly selective hydrogenation of acetylene. In this system, as authors reported, the Al sites weakly adsorb acetylene and catalyze the subsequent hydrogenation process, while the Ni sites facilitate hydrogen spillover for its subsequent hydrogenation. The resulting $\text{Al}_2\text{-Ni}_{\text{INC}}/\text{NCNT}$ catalyst demonstrated excellent selectivity toward ethylene, maintaining approximately 90% selectivity and exhibiting outstanding long-term stability without noticeable activity loss after 100 hours of continuous operation. Compared to strategies in previous studies focusing on the modifications in single-atom catalysts (SACs), nanoparticle systems, or support engineering (Chem. Rev. 2020, 120, 683-733; JACS 2023, 145, 26728-26735), this work presents a novel intergrade strategy through combining Al_2 and Ni_{INC} in carbon supports, significantly suppress the over-hydrogenation to ethane. Notably, the strategy is compelling due to its effective integration of distinct active sites, rather than relying on precise structural or electronic modulation. Based on this, I suggested this work can be considered for acceptance, but I still have several questions on this work for further revision as follows.

Comment 1:

In this work, I primarily interested in why the acetylene can adsorbed on Al sites for subsequent hydrogenation, rather than Ni sites. As we know, the Ni sites can enable strong adsorption compared to Al sites, which would appear to be more energetically favorable for acetylene adsorption.

Reply:

We gratefully acknowledge the reviewer for their insightful questions regarding the active sites. As correctly pointed out, Ni sites exhibit stronger adsorption for $^*\text{C}_2\text{H}_2$ and $^*\text{H}$ compared to Al sites (Figure 5c), resulting in preferential adsorption of C_2H_2 and H_2 on Ni sites. However, catalytic performance is governed not only by the

adsorption of reactants but also by the free-energy change associated with the rate-determining step (ΔG_{RDS} , namely activity) and the selectivity toward C_2H_4 and C_2H_6 . As calculated (Figure S35a), the ΔG_{RDS} value on Al sites is lower than Ni sites. Furthermore, $^*\text{C}_2\text{H}_4$ hydrogenation is thermodynamically less favorable compared to $^*\text{C}_2\text{H}_4$ desorption on Al sites, whereas it is more facile on Ni sites (Figure S35b). These thermodynamic analyses indicate that Al sites facilitate more favorable catalytic kinetics for the selective hydrogenation of acetylene, which is further supported by our calculated turnover frequency (TOF) values (Figure S36). Therefore, despite weaker adsorption of reactants, the superior intrinsic activity and enhanced selectivity of Al sites suggest that they serve as the primary active centers for acetylene selective hydrogenation.

To gain deeper insight into the adsorption behavior at Ni sites, we calculated the adsorption energies of $^*\text{C}_2\text{H}_2$ and $^*\text{H}$ under various $^*\text{C}_2\text{H}_2$ coverages (Figure S37-S39). As the coverage increases from 0.125 to 0.875 monolayer (ML), the adsorption energy of $^*\text{C}_2\text{H}_2$ shifts from initial -3 eV to +0.2 eV, indicating progressive surface saturation with adsorbed $^*\text{C}_2\text{H}_2$ species. In contrast, although $^*\text{H}$ adsorption energy exhibits a slight increase with increasing coverage, it remains negative, implying that $^*\text{H}$ species can still be generated from H_2 dissociation. Therefore, we propose that while Ni sites exhibited strong adsorption towards $^*\text{C}_2\text{H}_2$ and $^*\text{H}$, the favorable $^*\text{C}_2\text{H}_2$ adsorption mainly leads to high coverage. In contrast, the readily formed $^*\text{H}$ species could migrate to adjacent Al sites, where they participate in the hydrogenation of $^*\text{C}_2\text{H}_2$.

Figure S35a. The free energy of the rate-determining step (RDS) on the Al₂ dual-atom site and Ni

(111) surface; **b**. The desorption energy and hydrogenation energy of the C_2H_4 molecule on the Al_2 dual-atom site and Ni (111) surface.

Figure S36. Logarithmic turnover frequencies (log (TOF)) of Ni (111) and Al_2 dual-atom sites at the reaction temperature of 430 K.

Figure S37. Adsorption energies of $*C_2H_2$ and $*H$ on the Ni (111) surface under different C_2H_2 coverages.

Figure S38. Atomic visualization of *C_2H_2 on the Ni (111) surface at different *C_2H_2 coverages.

Figure S39. Atomic visualization of *H on the Ni (111) surface at different *C_2H_2 coverages.

Revisions made in the main text:

Indeed, the Ni sites exhibit a significant increase in free energy during the *CH_2CH_2 desorption step compared to its hydrogenation (Figure 5d and S33) regardless of whether the systems involve clusters or bulk catalysts. Specifically, the desorption energy of *CH_2CH_2 on Ni sites of all sizes is markedly high, indicating strong adsorption. The subsequent hydrogenation of *CH_2CH_2 readily occurs on both Ni clusters of various sizes and Ni (111) surface, with reaction energies ranging from -0.99 to -0.48 eV. This implies that the multi-bridged adsorption configuration stabilizes the *CH_2CH_2 intermediate, rendering desorption energetically unfavorable and leading to poor ethylene selectivity. These above results are in good agreement with DRIFT analysis. Furthermore, the reaction energy for hydrogen dissociation has been calculated (Figure 5c and S34). Ni species display a stronger thermodynamic propensity for hydrogen activation than the Al₂ site, which is energetically less favorable, demonstrating that the Ni (111) surface is highly effective at activating H₂. Thus, Ni sites exhibit stronger adsorption for C₂H₂ and H₂ compared to Al sites. However, catalytic performance is governed not only by the adsorption of reactants but also by the free-energy change associated with the rate-determining step (ΔG_{RDS} , namely activity) and the selectivity toward C₂H₄ and C₂H₆. As calculated (Figure S35a), the ΔG_{RDS} value on Al sites is lower than Ni sites. Furthermore, *C_2H_4 hydrogenation is thermodynamically less favorable compared to *C_2H_4 desorption on Al sites, whereas it is more facile on Ni sites (Figure S35b). These thermodynamic analyses indicate that Al sites facilitate more favorable catalytic kinetics for the

selective hydrogenation of acetylene, which is further supported by our calculated turnover frequency (TOF) values (Figure S36). Therefore, despite weaker adsorption of reactants, the superior intrinsic activity and enhanced selectivity of Al sites suggest that they serve as the primary active centers for acetylene selective hydrogenation.

To gain deeper insight into the adsorption behavior at Ni sites, the adsorption energies of $*C_2H_2$ and $*H$ under various $*C_2H_2$ coverages were calculated (Figure S37-S39). As the coverage increases from 0.125 to 0.875 monolayer (ML), the adsorption energy of $*C_2H_2$ shifts from initial -3 eV to +0.2 eV, indicating progressive surface saturation with adsorbed $*C_2H_2$ species. In contrast, although $*H$ adsorption energy exhibits a slight increase with increasing coverage, it remains negative, implying that $*H$ species can still be generated from H_2 dissociation. Therefore, we propose that while Ni sites exhibited strong adsorption towards $*C_2H_2$ and $*H$, the favorable $*C_2H_2$ adsorption mainly leads to high coverage. In contrast, the readily formed $*H$ species could migrate to adjacent Al sites, where they participate in the hydrogenation of $*C_2H_2$. Based on the results from *in situ* spectroscopic characterizations and theoretical calculations, it could be concluded that the construction of the synergistic Ni nanoclusters and Al_2 active centers offers excellent activity and selectivity. In this system, nickel species facilitate H_2 dissociation, while the Al_2 sites promote the spillover of active H^* species, enabling their reaction with π -bound acetylene and suppressing ethylene desorption, thereby inhibiting over-hydrogenation and oligomerization.

Comment 2:

Figure 5 presents DFT calculated results, including reaction energies, adsorption energies, and notably the D value, which is important for elucidating the underlying mechanism. Given that these results serve different roles in illustrating the reaction mechanism, detailed definitions and corresponding descriptions should be provided in the main text to aid reader comprehension.

Reply:

We appreciate the reviewer's suggestion on computational details. As the reviewer

advised, we added the related description in the main text regarding definitions of D values, reaction energy, adsorption as follows.

Revisions made in the Supporting Information:

The free energy of all species involved in acetylene selective hydrogenation is calculated below:

$$G = E + E_{zpe} + PV - TS$$

Where G represents Gibbs free energy, while E and E_{zpe} are the electronic energy via DFT calculations and zero-point energies, respectively. The pressure-volume (PV) contribution is considered negligible. The temperature (T) is considered as 298 K, and S is the entropy.

The dissociation energy of H_2 is calculated in the following equation:

$$E_{dis} = E_{*2H} - E_* - E_{H_2}$$

E_{dis} represents the dissociation energy, where E_{*2H} is the total energy of the catalyst after adsorbing two H atoms, E_* is the energy of the clean catalyst surface, and E_{H_2} is the energy of the gas-phase H_2 molecule. This energy reflects the difficulty of H_2 dissociation on the catalyst surface. a more negative value indicates that H_2 molecules dissociate more easily on the surface.

The desorption energy of C_2H_4 (E_{des}) is calculated below:

$$E_{des} = E_* + E_{C_2H_4} - E_{*C_2H_4}$$

Here, $E_{C_2H_4}$ represents the energy of the C_2H_4 molecule in the gas phase, and $E_{*C_2H_4}$ denotes the total energy of the catalyst surface with an adsorbed C_2H_4 molecule.

$E_{desorption}$ represents the energy required for the desorption of C_2H_4 from the catalyst surface; a more negative value of $E_{desorption}$ indicates that C_2H_4 can desorb more easily from the surface.

The hydrogenation energy of C_2H_4 (E_{hyd}) is calculated below:

$$E_{hyd} = E_{*C_2H_5} - E_{*C_2H_4} - \frac{1}{2}E_{H_2}$$

Where $E_{*C_2H_5}$ represents the total energy of the catalyst surface with an adsorbed C_2H_5 species, and $E_{hydrogenation}$ denotes the energy required for the hydrogenation of C_2H_4 to form C_2H_5 . A more negative $E_{hydrogenation}$ value indicates that the hydrogenation of C_2H_4 to C_2H_5 is more favourable.

The definition of the D-value is as follow:

$$D \text{ value} = E_{desorption} - E_{hydrogenation}$$

The more negative this value is, the easier the catalyst is to remove the C_2H_4 molecules, and the more difficult hydrogenation occurs.

Comment 3:

Figure 5 provides thermodynamic results to explore the role of Ni and Al sites in hydrogenation of acetylene, but it is not clear to describe particularly given the energy values dependent on DFT methodologies. I would recommend including additional electronic structure analyses, which could provide more direct insight into the interaction between acetylene and the active sites.

Reply:

We appreciate the reviewer's insightful comments. The differential charge density plots presented in Figure S32 clearly reveal the distinct adsorption behaviors of C_2H_2 and C_2H_4 on the Al_2 site and Ni (111) surface. On Al_2 site, charge redistribution is predominantly localized between the C-C bond and the surface, indicating a weak interaction. In the case of C_2H_4 , only slight surface polarization is observed, suggesting limited electronic coupling and facile product desorption. In contrast, adsorption of both C_2H_2 and C_2H_4 on Ni (111) surface induces pronounced charge transfer, wherein the C π electrons exhibit strong interactions with Ni d orbitals, resulting in significant charge redistribution and stronger binding between the adsorbate and the surface. Consequently, the weak adsorption on Al_2 facilitates the formation and release of C_2H_4 , whereas the strong chemisorption on Ni (111) hinders C_2H_4 desorption and reduces the selectivity for C_2H_4 hydrogenation.

Figure S32. Differential charge density of C_2H_2 and C_2H_4 adsorbed on (a) Al_2 and (b) Ni (111). Yellow and cyan regions denote electron accumulation and depletion, respectively.

Revisions made in the main text:

As expected, the generated $*CH_2CH_2$ intermediate adsorbs on Al_2 sites in a π -adsorbed configuration, which facilitates the desorption step with an energy of 0.70 eV. Indeed, this desorption energy is lower than that of further hydrogenation (referred to as the "D-value", which is negative in Figure 5c), suggesting that acetylene hydrogenation favors ethylene rather than ethane. However, acetylene adsorption on Ni sites occurs in a multi-bridged configuration with a strong binding energy of -2.88 eV, which favors conversion to ethane (namely the D-value is positive in Figure 5c). Notably, the differential charge density plots in Figure S32 clearly reveal the distinct adsorption behaviors of C_2H_2 and C_2H_4 on the Al_2 site and Ni (111) surface. On Al_2 site, the charge redistribution is predominantly localized between the

C-C bond and the surface, indicating a weak interaction. In the case of C₂H₄, only slight surface polarization is observed, suggesting limited electronic coupling and facile product desorption. In contrast, adsorption of both C₂H₂ and C₂H₄ on Ni (111) surface induce pronounced charge transfer, wherein the C π electrons strongly interact with Ni d orbitals, resulting in significant charge redistribution and stronger binding between the adsorbate and the surface. Consequently, the weak adsorption on Al₂ facilitates the formation and release of C₂H₄, whereas the strong chemisorption on Ni (111) hinders C₂H₄ desorption and reduces the selectivity for C₂H₄ hydrogenation.

Comment 4:

The atom labeling in the visualizations of models appears to be missing or insufficient. Clear labeling of key atoms is crucial for readers.

Reply:

We thank the reviewer for this valuable comment. The atom labeling to the related figure caption for clarity has been added.

Revisions made in the main text:

Figure 1. Synthesis principles and catalytic properties. (a) Schematic illustration of the synthesis of the Al₂-Ni_{INC}/NCNT synergistic catalyst (grey: C; blue: N; yellow: Al; light blue: Ni); (b) XRD patterns; Aberration-corrected HAADF-STEM images of (c) Al₂/NCNT, (d) Al₂-Ni_{INC}/NCNT and (e) Ni_{INC}/NCNT.

Figure 2. Analysis of electronic structure and coordination environments. (a) Al and (b) Ni K-edge XANES spectra; Fourier-transformed (c) Al and (d) Ni K-edge EXAFS spectra; (e) Al and (f) Ni K-edge EXAFS fitting in R space of Al₂-Ni_{INC}/NCNT (grey: C; blue: N; pink: Al; blue grey: Ni); (g) Al K-edge wavelet transform (WT)-EXAFS plots of Al foil and Al₂-Ni_{INC}/NCNT; (h) Ni K-edge wavelet transform (WT)-EXAFS plots of Ni foil and Al₂-Ni_{INC}/NCNT.

Figure 5. Reaction mechanism of acetylene hydrogenation. (a) Free energy profile of selective acetylene hydrogenation on Al₂ with Ni nanoclusters as contrast; (b) Configuration of reaction intermediates (grey: C; blue: N; pink: Al; white: H); (c) Energies of hydrogen dissociation and D-value standing for energy difference

between C₂H₄ desorption and its further hydrogenation; (d) Reaction energy for ethylene desorption and hydrogenation on various Ni sites on Ni₁₉, Ni₃₄ and Ni₅₂ and Ni (111).

Comment 5:

As author calculated, the behavior on various scale of Ni models were studied to understand the role of Ni sites. But the size of catalysts can affect the catalysis, which is commonly reported (*J. Catal.* 2023, 425, 70-79). Therefore, the discussion on particle size effect should be considered, whether the size effect is not affected the results, especially the H₂ dissociation and ethylene adsorption.

Reply:

We sincerely thank for the reviewer's comments on particle size effect. The particle size effect is important for catalysis as the activities of catalytic sites are related to the coordinated environment (*Science* 2015, 350,185; *J. Catal.* 2023, 425, 70). As shown in Figure S31 and 5d, Ni clusters of different sizes were constructed and analyzed, revealing that the adsorption energy varies with the size. The adsorption energy of *C₂H₄ remains notably low, indicating strong adsorption on Ni sites (Figure S33). In contrast, the subsequent hydrogenation of *C₂H₄ readily occurs on both Ni clusters of various sizes and the Ni (111) surface, with reaction energies ranging from -0.99 to -0.48 eV. These results suggest that, for this reaction, Ni sites lead to over-hydrogenation rather than the selective formation of ethylene.

Revisions made in the main text:

Indeed, the Ni sites exhibit a significant increase in free energy during the *CH₂CH₂ desorption step compared to its hydrogenation (Figure 5d and S33) regardless of whether the systems involve clusters or bulk catalysts. Specifically, the desorption energy of *CH₂CH₂ on Ni sites of all sizes is markedly high, indicating strong adsorption. The subsequent hydrogenation of *CH₂CH₂ readily occurs on both Ni clusters of various sizes and Ni (111) surface, with reaction energies ranging from -0.99 to -0.48 eV. This implies that the multi-bridged adsorption configuration stabilizes the *CH₂CH₂ intermediate, rendering desorption energetically unfavorable

and leading to poor ethylene selectivity.

Comment 6:

Compared to Al_2O_3 , Al_2/NCNT and Al_1/NCNT have significantly improved activity. It is recommended to supplement the performance experiment of NCNT to rule out the interference of the support on atomically precise Al catalysts.

Reply:

Thanks for your constructive comment. The catalytic performance of NCNT support was evaluated in the absence of any metal specie. As shown in Figure S14, the NCNT support exhibited negligible catalytic activity under the same conditions.

Figure S14. Acetylene conversion of the NCNT.

Revisions made in the main text:

Remarkably, atomic-scale Al sites exhibit excellent activation ability of acetylene on the premise of excluding the interference of NCNT support (Figure S14) and reactor system (Figure S15), with Al_2/NCNT displaying better catalytic performance compared to single-atom Al counterpart (Figure 3a).

Comment 7:

The conversion should be controlled below 15% for Arrhenius plot measurement. This is critical to validate the reliability of the activation energy and reaction order. The authors should clarify this point. Meanwhile, the authors claim that atomically precise Al catalysts exhibit good activity for the acetylene hydrogenation, but missing

the sufficient kinetic analysis, especially for Al₁ catalysts, which is important for catalytic mechanism.

Reply:

We appreciate the reviewer's insightful comments. Indeed, the previously calculated activation energies and reaction orders were determined under conditions limited below 15% conversion, which have been replenished in the revised manuscript.

The activation energy, reaction orders of acetylene and hydrogen on Al₁/NCNT have been supplemented in Figure 3e-g as below.

Figure 3. Catalytic performance of acetylene hydrogenation. (e) Arrhenius plots; Reaction order of (f) acetylene and (g) hydrogen, respectively.

Revisions made in the main text:

To clarify the kinetics advantages of the Al₂-Ni₁NC/NCNT synergetic catalyst, the intrinsic activity, characterized by apparent activation energy (E_a) and reaction order was further analyzed under conditions below 15% conversion to eliminate the influence of mass or heat transfer. Firstly, kinetic measurements using the Arrhenius equation further affirm the intrinsic activity trend, with Al₂-Ni₁NC exhibiting a competitive advantage through a lower activation energy of 55.0 kJ mol⁻¹ (Figure 3e). The reaction order (n) for acetylene was also determined by plotting intrinsic activity versus pressure (Figure 3f), yielding a value of 0.19 for Al dual-atom sites integrated with Ni nanoclusters, while a negative value (-0.39) was observed for the aggregated Ni species. This negative reaction order suggests that high acetylene coverage inhibits the reaction rate due to strong adsorption. Additionally, the hydrogen reaction order, shown in Figure 3g, exhibits a linear relationship, with values of 0.39 for Al₂-Ni₁NC/NCNT and 1.07 for Ni₁NC/NCNT. The near-zero reaction order indicates that

the catalytic activity of Al₂-Ni_{INC} is determined by the structure rather than hydrogen pressure. This implies that the synergy between Al dual-atom sites and Ni nanoclusters effectively reduces competitive acetylene adsorption while enhancing hydrogen activation and dissociation.

Reviewer #2:

The authors of this manuscript propose an intriguing concept, where atomically dispersed Al sites (in pairs) work in concert with Ni nanoclusters to achieve high selectivity for acetylene hydrogenation, an industrially relevant reaction. The use of atomically dispersed Al on carbon nanotubes is novel, however I would like to know if the catalyst will survive any oxidation which may be required to burn off the oligomers in an industrial setting.

Reply:

We sincerely appreciate your helpful comment. As suggested, we regenerated the used Al₂-Ni_{INC}/NCNT catalyst by oxidative treatment to remove oligomers, following a protocol adapted from industrial practices used at Jam domestic petrochemical for acetylene hydrogenation (*J. Nat. Gas Sci. Eng.* 2016, 34, 1382; *Chem. Eng. J.* 2012, 198-199, 491). Firstly, a nitrogen stream at a flow rate of 100 mL min⁻¹ was fed to the reactor at 115 °C for 60 min. The temperature was then increased to 165 °C while maintaining the nitrogen flow to stabilize the catalyst bed. In the next step, in addition to nitrogen, water vapor (H₂O (g)) at a flow rate of 0.02 mL min⁻¹ was injected as the temperature was raised from 165 to 400 °C, and this temperature was maintained for 90 min to wash out and reform light hydrocarbons. Subsequently, the heavy hydrocarbons prone to coking were burnt by introducing oxygen with the flow rate gradually increased from 5 to 40 mL min⁻¹ at 500 °C. Following this step, nitrogen was purged through the reactor for 130 min to displace residual oxygen. Then, a hydrogen stream at 10 mL min⁻¹ was introduced to reduce the oxidized nickel species back to metallic Ni. Finally, nitrogen was passed through the system until the bed temperature decreased to ambient conditions, yielding the regenerated Al₂-Ni_{INC}/NCNT catalyst.

Raman and XAS results in the figure below demonstrate that the industrial regeneration process could effectively remove the oligomers and coking from the surface of $\text{Al}_2\text{-Ni}_{\text{NC}}/\text{NCNT}$ catalyst, while the atomically dispersed Al sites and Ni nanoclusters remain intact. This is further supported by the catalytic performance of the regenerated $\text{Al}_2\text{-Ni}_{\text{NC}}/\text{NCNT}$ catalyst, which is basically consistent with that of the corresponding fresh catalyst.

Figure Raman spectra of the regenerated $\text{Al}_2\text{-Ni}_{\text{NC}}/\text{NCNT}$.

Figure S22 Fourier-transformed (a) Al and (b) Ni K-edge EXAFS spectra with fitting of spent $\text{Al}_2\text{-Ni}_{\text{NC}}/\text{NCNT}$ after regeneration.

Figure Acetylene conversion and ethylene selectivity over fresh and regenerated Al₂-Ni_{NC}/NCNT catalysts.

Revisions made in the main text:

Notably, Al₂-Ni_{NC}/NCNT demonstrates exceptional thermal and chemical stability under hydrogenation conditions, maintaining both activity and selectivity for at least 150 h at 142 °C, with no visible decline even upon regeneration (Figure 3h).

Revisions made in the Supporting Information:

The used Al₂-Ni_{NC}/NCNT catalyst by oxidative treatment to remove oligomers, following a protocol adapted from industrial practices used at Jam domestic petrochemical for acetylene hydrogenation (*J. Nat. Gas Sci. Eng.* 2016, 34, 1382; *Chem. Eng. J.* 2012, 198-199, 491). Firstly, a nitrogen stream at a flow rate of 100 mL min⁻¹ was fed to the reactor at 115 °C for 60 min. The temperature was then increased to 165 °C while maintaining the nitrogen flow to stabilize the catalyst bed. In the next step, in addition to nitrogen, water vapor (H₂O (g)) at a flow rate of 0.02 mL min⁻¹ was injected as the temperature was raised from 165 to 400 °C, and this temperature was maintained for 90 min to wash out and reform light hydrocarbons. Subsequently, the heavy hydrocarbons prone to coking were burnt by introducing oxygen with the flow rate gradually increased from 5 to 40 mL min⁻¹ at 500 °C. Following this step, nitrogen was purged through the reactor for 130 min to displace residual oxygen. Then, a hydrogen stream at 10 mL min⁻¹ was introduced to reduce the oxidized nickel species back to metallic Ni. Finally, nitrogen was passed through the system until the

bed temperature decreased to ambient conditions, yielding the regenerated Al₂-Ni_{NC}/NCNT catalyst.

Also, I have a few concerns about the manuscript which should be addressed by the authors:

Comment 1:

Metallic Ni is not a selective catalyst for this reaction and leads to over hydrogenation. So, if you have any metallic clusters exposed, they will lower the selectivity of the reaction. How do you keep the Ni from doing the hydrogenation (non-selectively) since it is a more active catalyst than the Al. From the schematic diagram of the synthesis, in Figure 1, it appears that the Ni is added after the addition of the Al. So, the Ni is exposed to the gas phase and can potentially catalyze the non-selective over hydrogenation to ethane.

Reply:

We thank the reviewer for their insightful comments regarding the potential non-selective hydrogenation behavior of Ni clusters. To address this, we performed calculations of *C₂H₂ adsorption on Ni surfaces under different coverages, as shown in Figure S37-S39. To further elucidate the strong adsorption of Ni sites, we calculated the adsorption energies of *C₂H₂ and *H at various *C₂H₂ coverages. As the *C₂H₂ coverage increases from 0.125 to 0.875 ML, the adsorption energy of *C₂H₂ rises from -3.0 eV to 0.2 eV, indicating that the surface could be covered by *C₂H₂ species. In addition, although the adsorption energy of *H slightly increases with coverage, it remains negative, suggesting that *H species could still be generated from H₂ dissociation. Therefore, we infer that while Ni sites exhibit strong adsorption toward *C₂H₂ and *H, the favorable *C₂H₂ adsorption mainly leads to high surface coverage. In contrast, the readily generated *H species could migrate to adjacent Al sites, where *C₂H₂ hydrogenation preferentially occurs. This mechanism ensures that the Ni surface does not dominate the non-selective hydrogenation process.

Figure S37. Adsorption energies of C₂H₂ and H on the Ni (111) surface under different C₂H₂ coverages.

Figure S38. Atomic visualization of *C₂H₂ on the Ni (111) surface at different *C₂H₂ coverages.

Figure S39. Atomic visualization of *H on the Ni (111) surface at different *C₂H₂ coverages.

Revisions made in the main text:

Indeed, the Ni sites exhibit a significant increase in free energy during the *CH₂CH₂ desorption step compared to its hydrogenation (Figure 5d and S33) regardless of whether the systems involve clusters or bulk catalysts. Specifically, the desorption energy of *CH₂CH₂ on Ni sites of all sizes is markedly high, indicating strong adsorption. The subsequent hydrogenation of *CH₂CH₂ readily occurs on both Ni clusters of various sizes and Ni (111) surface, with reaction energies ranging from

-0.99 to -0.48 eV. This implies that the multi-bridged adsorption configuration stabilizes the *CH_2CH_2 intermediate, rendering desorption energetically unfavorable and leading to poor ethylene selectivity. These above results are in good agreement with DRIFT analysis. Furthermore, the reaction energy for hydrogen dissociation has been calculated (Figure 5c and S34). Ni species display a stronger thermodynamic propensity for hydrogen activation than the Al₂ site, which is energetically less favorable, demonstrating that the Ni (111) surface is highly effective at activating H₂. Thus, Ni sites exhibit stronger adsorption for C₂H₂ and H₂ compared to Al sites. However, catalytic performance is governed not only by the adsorption of reactants but also by the free-energy change associated with the rate-determining step (ΔG_{RDS} , namely activity) and the selectivity toward C₂H₄ and C₂H₆. As calculated (Figure S35a), the ΔG_{RDS} value on Al sites is lower than Ni sites. Furthermore, *C_2H_4 hydrogenation is thermodynamically less favorable compared to *C_2H_4 desorption on Al sites, whereas it is more facile on Ni sites (Figure S35b). These thermodynamic analyses indicate that Al sites facilitate more favorable catalytic kinetics for the selective hydrogenation of acetylene, which is further supported by our calculated turnover frequency (TOF) values (Figure S36). Therefore, despite weaker adsorption of reactants, the superior intrinsic activity and enhanced selectivity of Al sites suggest that they serve as the primary active centers for acetylene selective hydrogenation.

To gain deeper insight into the adsorption behavior at Ni sites, the adsorption energies of *C_2H_2 and *H under various *C_2H_2 coverages were calculated (Figure S37-S39). As the coverage increases from 0.125 to 0.875 monolayer (ML), the adsorption energy of *C_2H_2 shifts from initial -3 eV to +0.2 eV, indicating progressive surface saturation with adsorbed *C_2H_2 species. In contrast, although *H adsorption energy exhibits a slight increase with increasing coverage, it remains negative, implying that *H species can still be generated from H₂ dissociation. Therefore, we propose that while Ni sites exhibited strong adsorption towards *C_2H_2 and *H , the favorable *C_2H_2 adsorption mainly leads to high coverage. In contrast, the readily formed *H species could migrate to adjacent Al sites, where they participate in the hydrogenation of *C_2H_2 . Based on the results from *in situ* spectroscopic

characterizations and theoretical calculations, it could be concluded that the construction of the synergistic Ni nanoclusters and Al₂ active centers offers excellent activity and selectivity. In this system, nickel species facilitate H₂ dissociation, while the Al₂ sites promote the spillover of active H* species, enabling their reaction with π -bound acetylene and suppressing ethylene desorption, thereby inhibiting over-hydrogenation and oligomerization.

Comment 2:

The AC-STEM images need revision.

a. First, Figure 1c is too dark. It needs to be brightened to be able to see clearly. Also, a corresponding image without the circles needs to be included side by side with Figure 1c, so the reader can see clearly the basis for choosing to circle the atoms that are being highlighted.

b. Next, the authors circle some bright objects on the surface of the Ni nanoparticles in Figure 1d. As in comment 1a above, the authors need to show in the supporting information this image along with one without the circles. And they need to explain the basis for concluding there are Al atoms on the surface of the Ni. Al is much lighter than Ni, furthermore similar bright objects are also seen on the Ni catalyst (Figure 1e). So, without some other evidence, perhaps XPS, or LEIS and also EDS, we cannot be sure that the Ni is indeed covered by the Al atoms.

Reply:

We sincerely appreciate your helpful comment. Figure 1c has been brightened to be able to see clearly. A corresponding image without the circles has been provided in Figure S5, so the reader could see clearly the basis for choosing to circle the atoms that are being highlighted. Meanwhile, the original version of Figure 1d without the circles has been added in Figure S5. As shown in EDS images presented in Figure S6, Al atoms are well distributed on the surrounding of Ni species.

Figure 1 Aberration-corrected HAADF-STEM images of (c) Al_2/NCNT (d) $\text{Al}_2\text{-Ni}_{\text{NC}}/\text{NCNT}$ (e) Al_1/NCNT .

Figure S5 Aberration-corrected HAADF-STEM images of (a) Al_2/NCNT (b) $\text{Al}_2\text{-Ni}_{\text{NC}}/\text{NCNT}$ (c) Al_1/NCNT without the circles and (d) $\text{Ni}_{\text{NC}}/\text{NCNT}$.

Figure S6 The EDX element maps of $\text{Al}_2\text{-Ni}_{\text{NC}}/\text{NCNT}$ catalyst.

Revisions made in the main text:

Additionally, it is observed that while the Al distribution remains uniform, the Ni nanoclusters are segregated and surrounded by abundant Al dual-atom in $\text{Al}_2\text{-Ni}_{\text{NC}}/\text{NCNT}$, as confirmed by the HAADF-STEM and EDS image (Figure 1d and S6).

Comment 3:

The analysis of the EXAFS via fitting the data does not show any evidence for Ni nearest neighbors to the Al, or vice versa. But this is implied in Figure 1d. Do the Ni clusters expose metallic Ni atoms to the gas phase? If so, what stops them from doing the non-selective hydrogenation of ethylene to ethane. Are the Ni clusters physically distant from the Al atoms on the carbon.

Reply:

Thanks for your constructive comment. Based on EDS results (Figure S6), the Ni clusters are surrounded by the Al atoms. However, no chemical bonding is observed between Al atoms and Ni species, as evidenced by the absence of Ni-Al bonding in the EXAFS analysis based on data fitting.

Indeed, Ni clusters expose metallic Ni atoms to the gas phase. When Ni clusters are surrounded by the Al atoms, the ΔG_{RDS} value, representing the free energy change of the rate-determining step, on Al sites is lower than that on Ni sites (Figure S35a). Moreover, compared to $^*\text{C}_2\text{H}_4$ desorption, $^*\text{C}_2\text{H}_4$ hydrogenation is less favorable on

Al sites but more facile on Ni sites (Figure S35b). These thermodynamic results indicate that Al sites exhibit more favorable catalytic kinetics for the selective hydrogenation of acetylene, which is further supported by our calculated turnover frequency (TOF) values (Figure S36). Therefore, despite the stronger adsorption on Ni sites, the higher activity observed on Al sites suggests that they serve as the primary active centers for acetylene selective hydrogenation.

Figure S6 The EDX element mapping of $\text{Al}_2\text{-Ni}_{\text{NC}}/\text{NCNT}$ catalyst.

Figure S35a. The free energy of the rate-determining step (RDS) on the Al_2 dual-atom site and Ni (111) surface; **b.** The desorption energy and hydrogenation energy of the C_2H_4 molecule on the Al_2 dual-atom site and Ni (111) surface.

Figure S36. Logarithmic turnover frequencies (log (TOF)) of Ni (111) and Al₂ dual-atom sites at the reaction temperature of 430 K.

Comment 4:

There are several aspects of the catalytic measurements that need clarification:

- a. It is stated that the Al₂/NCNT catalyst shows no influence of temperature on the conversion or selectivity. Does the reactor system have any background reactivity? It seems odd that there is.

Reply:

Thanks for your question. The performance evaluation results of the reactor system have been presented in Figure S15. In the absence of any catalysts, the reactor system exhibits no reactivity. Regarding the increase in reaction rate with temperature over the Al₂/NCNT catalyst, we apologize for the previous description, which may have caused confusion; this has been revised accordingly in the updated manuscript.

Figure S15 Reactivity of reactor system.

Revisions made in the main text:

Remarkably, atomic-scale Al sites exhibit excellent activation ability of acetylene on the premise of excluding the interference of NCNT support (Figure S14) and reactor system (Figure S15), with Al₂/NCNT displaying better catalytic performance compared to single-atom Al counterpart (Figure 3a). This behavior markedly contrasts with the inert α -Al₂O₃ and Al powders, which is usually used as substrate to load Pd catalysts for C₂ selective hydrogenation in the industry. However, higher acetylene conversion for Al₂/NCNT catalysts requires the elevated reaction temperature (>240 °C in Figure 3a and S16) although exhibiting excellent ethylene selectivity (94.0%).

b. In the supporting information, they need to show more details of the reaction products, for instance the concentrations of acetylene, ethylene and ethane as a function of temperature, or a function of space velocity, so the reader can see when the formation of ethane starts and what happens as acetylene is fully consumed. Figure 3 only shows conversion and selectivity. It is important to see all the products and to understand how much ethane is being produced.

Reply:

We sincerely appreciate your helpful comments. The concentrations of acetylene, ethylene, ethane and oligomers as a function of temperature have been supplemented in Figure S17-S18. It is found that ethane starts to form over Al₂-Ni_{INC}/NCNT catalyst when the temperature increases to 88 °C. When acetylene is fully consumed, the ethane is further generated; however, the corresponding amount is significantly lower than that of Ni_{INC}/NCNT catalyst.

Based on your suggestion, the selectivity of all products have been presented in Figure 3b-d. As expected, Al₂-Ni_{INC}/NCNT significantly enhances catalytic activity, achieving 99.98% conversion at 157 °C, far superior to Al₂/NCNT (49.75% @ 340 °C) and Ni_{INC}/NCNT (99.93% @ 200 °C) (Figure 3a and Figure S17-S18). More importantly, even at complete acetylene conversion, Al₂-Ni_{INC}/NCNT maintains ethylene selectivity of 90.26% (Figure 3b), with only minor formation of ethane

(6.17% of selectivity) and oligomers (3.57% of selectivity) in Figure 3c-d.

Figure S17. The concentrations of (a) acetylene, (b) ethylene (c) ethane and (d) oligomers as a function of temperature over Al₂-Ni_{NC}/NCNT.

Figure S18. The concentrations of (a) acetylene, (b) ethylene (c) ethane and (d) oligomers as a function of temperature over Ni_{NC}/NCNT.

Figure 3. Catalytic performance of acetylene hydrogenation. (a) Conversion as a function of reaction temperature; Selectivity of (b) C_2H_4 , (c) C_2H_6 and (d) oligomers as a function of reaction temperature; (e) Arrhenius plots; (f, g) Reaction order of acetylene and hydrogen, respectively; (h) Durability test on $Al_2-Ni_{NC}/NCNT$ at $142\text{ }^\circ C$. (Reaction condition: a hydrogen to acetylene ratio of 20:1; space velocity = $9000\text{ mL g}^{-1}\text{ h}^{-1}$; atmospheric pressure. Regeneration method is originated from the Jam domestic petrochemical for acetylene hydrogenation^{38,39}).

Revisions made in the main text:

The catalytic conversion and selectivity of Al_1 - and Al_2 - based samples supported on NCNT were first evaluated in acetylene selective hydrogenation, with the concentration of reactants and products as a function of temperature shown in Figure 3.

As expected, $Al_2-Ni_{NC}/NCNT$ significantly enhances catalytic activity, achieving 99.98% conversion at $157\text{ }^\circ C$, far superior to $Al_2/NCNT$ (49.75% @ $340\text{ }^\circ C$) and $Ni_{NC}/NCNT$ (99.93% @ $200\text{ }^\circ C$), as shown in Figure 3a and Figure S17-S18. More importantly, even at complete acetylene conversion, $Al_2-Ni_{NC}/NCNT$ maintains

ethylene selectivity of 90.26% (Figure 3b), with only minor formation of ethane (6.17% of selectivity) and oligomers (3.57% of selectivity), as depicted in Figure 3c-d. In contrast, the Ni_{INC} catalyst, which lacks adjacent Al dual-atom, exhibits markedly lower ethylene selectivity (43.70% at 99.93% conversion). Based on the temperature-dependent concentration profiles of acetylene, ethylene, ethane and oligomers (Figure S17-S18), it is found that ethane and oligomers start to form over Al₂-Ni_{INC}/NCNT catalyst when the temperature increases to 88 °C; however, their corresponding amounts are significantly lower than that of Ni_{INC} catalyst. This difference may be attributed to the formation of ethylidene (=CHCH₃) on Ni nanoclusters, resulting in excessive hydrogenation to ethane and polymerization into oligomers (Figure 3c-3d).

c. Did they quantify the rate of formation of oligomers, or green oil during their long-term tests.

Reply:

We sincerely thank the reviewer for this comment. The oligomer formation rate was calculated at complete acetylene conversion (Figure S19), where Al₂-Ni_{INC}/NCNT exhibits an oligomers formation rate of 0.29 mol_{oligomers} mol_{Ni}⁻¹ h⁻¹. This is significantly lower than that of Ni_{INC}/NCNT (1.72 mol_{oligomers} mol_{Ni}⁻¹ h⁻¹), indicating that the construction of synergistic sites can effectively suppress the deposition of carbonaceous species.

Figure S19 The formation rates of oligomers over Ni_{INC}/NCNT and Al₂-Ni_{INC}/NCNT at complete acetylene conversion

Revisions made in the main text:

Meanwhile, the lower formation rate of oligomers of $0.29 \text{ mol}_{\text{oligomers}} \text{ mol}_{\text{Ni}}^{-1} \text{ h}^{-1}$ over $\text{Al}_2\text{-Ni}_{\text{NC}}/\text{NCNT}$ in Figure S19, compared to that of $\text{Ni}_{\text{NC}}/\text{NCNT}$ ($1.72 \text{ mol}_{\text{oligomers}} \text{ mol}_{\text{Ni}}^{-1} \text{ h}^{-1}$) indicates that the construction of synergistic sites could suppress the deposition of carbonaceous species.

Reviewer #3:

This manuscript reports the synthesis of Al dual-atom sites and Ni clusters (ca. 2.5 nm average size) on nitrogen-doped carbon nanotubes ($\text{Al}_2\text{-Ni}_{\text{NC}}\text{-NCNT}$) and its catalytic activity for the selective semi-hydrogenation reaction of acetylene in ethylene streams. The corresponding blanks ($\text{Al}_2\text{-NCNT}$, $\text{Ni}_{\text{NC}}\text{-NCNT}$ and also $\text{Al}\text{-NCNT}$ solid catalyst) are also prepared and tested. The reactivity shows that while $\text{Al}_2\text{-Ni}_{\text{NC}}\text{-NCNT}$ and $\text{Ni}_{\text{NC}}\text{-NCNT}$ are not far in catalytic activity (confirmed by the similar activation energy), the former is more selective, since apparently Al avoids over-hydrogenation of acetylene to ethane (and coupling reactions) on the Ni cluster sites. The latter is due to a much higher adsorption of acetylene on the Ni sites than in Al, favoring the undesired further reactions. If this is so, it means that Al overrules Ni when activating acetylene, and this is something that is not clear in the study. It could be that H_2 competes strongly with acetylene on Ni sites but, in this case, the catalytic activity of $\text{Ni}_{\text{NC}}\text{-NCNT}$ would be much lower, and it is not, just the selectivity.

Another point that makes the key claim of the study, i.e. that the symbiosis between dual Al sites (perform all the process except H_2 dissociation) and Ni (H_2 dissociation) triggers the reaction, to be doubtful is the absence of any significant characterization of the catalyst during or after reaction. The only data given are just a Raman, where any information on the metal sites cannot be obtained. The atomically-dispersed Al sites could be rapidly destroyed (aggregated, sublimated) under the thermal reaction conditions, even the Ni clusters could further aggregate or disperse. If the main advance of the paper is the originality of the catalytic sites and the mechanism, the nature and stability of the fresh catalyst during reaction must be clearly demonstrated.

With that said, other issues could be considered less important, but not for the high

standards of Nat. Commun.:

Reply:

We appreciate the reviewer's insightful comments. These thermodynamic analyses indicate that Al sites facilitate more favorable catalytic kinetics for the selective hydrogenation of acetylene, which is further supported by our calculated turnover frequency (TOF) values (Figure S36). Therefore, despite weaker adsorption of reactants, the superior intrinsic activity and enhanced selectivity of Al sites suggest that they serve as the primary active centers for acetylene selective hydrogenation.

To gain deeper insight into the adsorption behavior at Ni sites, we calculated the adsorption energies of *C_2H_2 and *H under various *C_2H_2 coverages (Figure S37-S39). As the coverage increases from 0.125 to 0.875 monolayer (ML), the adsorption energy of *C_2H_2 shifts from initial -3 eV to +0.2 eV, indicating progressive surface saturation with adsorbed *C_2H_2 species. In contrast, although *H adsorption energy exhibits a slight increase with increasing coverage, it remains negative, implying that *H species can still be generated from H_2 dissociation. Therefore, we propose that while Ni sites exhibited strong adsorption towards *C_2H_2 and *H , the favorable *C_2H_2 adsorption mainly leads to high coverage. In contrast, the readily formed *H species could migrate to adjacent Al sites, where they participate in the hydrogenation of *C_2H_2 .

Similar activation energies are observed for $Al_2-Ni_{INC}/NCNT$ and $Ni_{INC}/NCNT$ catalysts in the kinetic regime, specifically at low acetylene conversion (< 15%). However, the main challenge of selective hydrogenation of acetylene in C_2 streams lies in the strong competitive adsorption of acetylene, ethylene, and hydrogen at high conversion (particularly near complete conversion), which is governed not only by kinetics but also by thermodynamics. As expected, in this situation, the catalytic activity of $Ni_{INC}/NCNT$ is indeed lower, with complete acetylene conversion achieved at 200 °C compared to 157 °C for the Al_2-Ni_{INC} synergistic catalysts. Such a temperature gap is quite substantial in industrial process (*Appl. Surf. Sci.* 2022, 604, 154497).

Besides, additional and more comprehensive post-reaction characterizations have

been provided in response to the reviewer's comments, including XRD, XPS, and XAS spectra, as well as regeneration testing. As shown in Figure S21-S23, XRD, XAS and XPS results reveal that the atomically-dispersed Al sites and Ni clusters could be well maintained under the thermal reaction conditions. Consequently, Al₂-Ni_{INC}/NCNT catalyst also exhibits good stability in a long-period running, even regeneration.

Figure S36. Logarithmic turnover frequencies (log (TOF)) of Ni (111) and Al₂ dual-atom sites at the reaction temperature of 430 K.

Figure S37. Adsorption energies of C₂H₂ and H on the Ni (111) surface under different C₂H₂ coverages.

Figure S38. Atomic visualization of $*\text{C}_2\text{H}_2$ on the Ni (111) surface at different $*\text{C}_2\text{H}_2$ coverages.

Figure S39. Atomic visualization of $*\text{H}$ on the Ni (111) surface at different $*\text{C}_2\text{H}_2$ coverages.

Figure S21 XRD pattern of spent $\text{Al}_2\text{-Ni}_{\text{NC}}/\text{NCNT}$.

Figure S22 Fourier-transformed (a) Al and (b) Ni K-edge EXAFS spectra with fitting of spent $\text{Al}_2\text{-Ni}_{\text{NC}}/\text{NCNT}$.

Figure S23 Al 2p spectra of post-reaction $\text{Al}_2\text{-Ni}_{\text{NC}}/\text{NCNT}$.

Figure 3h. Durability test on $\text{Al}_2\text{-Ni}_{\text{NC}}/\text{NCNT}$ at 142 °C. (Reaction condition: a hydrogen to acetylene ratio of 20:1; space velocity = 9000 mL g⁻¹ h⁻¹; atmospheric pressure. Regeneration method is originated from the Jam domestic petrochemical for acetylene hydrogenation^{38,39}).

Revisions made in the main text:

Indeed, the Ni sites exhibit a significant increase in free energy during the

*CH₂CH₂ desorption step compared to its hydrogenation (Figure 5d and S33) regardless of whether the systems involve clusters or bulk catalysts. Specifically, the desorption energy of *CH₂CH₂ on Ni sites of all sizes is markedly high, indicating strong adsorption. The subsequent hydrogenation of *CH₂CH₂ readily occurs on both Ni clusters of various sizes and Ni (111) surface, with reaction energies ranging from -0.99 to -0.48 eV. This implies that the multi-bridged adsorption configuration stabilizes the *CH₂CH₂ intermediate, rendering desorption energetically unfavorable and leading to poor ethylene selectivity. These above results are in good agreement with DRIFT analysis. Furthermore, the reaction energy for hydrogen dissociation has been calculated (Figure 5c and S34). Ni species display a stronger thermodynamic propensity for hydrogen activation than the Al₂ site, which is energetically less favorable, demonstrating that the Ni (111) surface is highly effective at activating H₂. Thus, Ni sites exhibit stronger adsorption for C₂H₂ and H₂ compared to Al sites. However, catalytic performance is governed not only by the adsorption of reactants but also by the free-energy change associated with the rate-determining step (ΔG_{RDS} , namely activity) and the selectivity toward C₂H₄ and C₂H₆. As calculated (Figure S35a), the ΔG_{RDS} value on Al sites is lower than Ni sites. Furthermore, *C₂H₄ hydrogenation is thermodynamically less favorable compared to *C₂H₄ desorption on Al sites, whereas it is more facile on Ni sites (Figure S35b). These thermodynamic analyses indicate that Al sites facilitate more favorable catalytic kinetics for the selective hydrogenation of acetylene, which is further supported by our calculated turnover frequency (TOF) values (Figure S36). Therefore, despite weaker adsorption of reactants, the superior intrinsic activity and enhanced selectivity of Al sites suggest that they serve as the primary active centers for acetylene selective hydrogenation.

To gain deeper insight into the adsorption behavior at Ni sites, the adsorption energies of *C₂H₂ and *H under various *C₂H₂ coverages were calculated (Figure S37-S39). As the coverage increases from 0.125 to 0.875 monolayer (ML), the adsorption energy of *C₂H₂ shifts from initial -3 eV to +0.2 eV, indicating progressive surface saturation with adsorbed *C₂H₂ species. In contrast, although *H adsorption energy exhibits a slight increase with increasing coverage, it remains negative,

implying that *H species can still be generated from H₂ dissociation. Therefore, we propose that while Ni sites exhibited strong adsorption towards *C₂H₂ and *H, the favorable *C₂H₂ adsorption mainly leads to high coverage. In contrast, the readily formed *H species could migrate to adjacent Al sites, where they participate in the hydrogenation of *C₂H₂. Based on the results from *in situ* spectroscopic characterizations and theoretical calculations, it could be concluded that the construction of the synergistic Ni nanoclusters and Al₂ active centers offers excellent activity and selectivity. In this system, nickel species facilitate H₂ dissociation, while the Al₂ sites promote the spillover of active H* species, enabling their reaction with π -bound acetylene and suppressing ethylene desorption, thereby inhibiting over-hydrogenation and oligomerization.

Notably, Al₂-Ni_{INC}/NCNT demonstrates exceptional thermal and chemical stability under hydrogenation conditions, maintaining both activity and selectivity for at least 150 h at 142 °C, with no visible decline even upon regeneration (Figure 3h). To investigate the nature and stability of this catalyst during reaction, XRD pattern and XAS analysis have been provided. As shown in Figure S21-S22, the atomically-dispersed Al sites and Ni clusters of Al₂-Ni_{INC}/NCNT could be well maintained after the reaction. Meanwhile, Al 2p XPS results (Figure S23) further indicate that the average oxidation state of Al species well keeps between 0 and +3. That is to say, the atomically-dispersed Al sites and Ni clusters could be well maintained under the thermal reaction conditions, with excellent resistance to carbon deposition (Figure S24). In contrast, Ni_{INC}/NCNT shows a gradual decrease in ethylene selectivity, approximately with a loss of ~90% within the first five hours (Figure S25), and Raman measurements indicate severe coking (Figure S24).³⁹

Revisions made in the Supporting Information:

The used Al₂-Ni_{INC}/NCNT catalyst by oxidative treatment to remove oligomers, following a protocol adapted from industrial practices used at Jam domestic petrochemical for acetylene hydrogenation (*J. Nat. Gas Sci. Eng.* 2016, 34, 1382; *Chem. Eng. J.* 2012, 198-199, 491). Firstly, a nitrogen stream at a flow rate of 100 mL min⁻¹ was fed to the reactor at 115 °C for 60 min. The temperature was then increased

to 165 °C while maintaining the nitrogen flow to stabilize the catalyst bed. In the next step, in addition to nitrogen, water vapor (H₂O (g)) at a flow rate of 0.02 mL min⁻¹ was injected as the temperature was raised from 165 to 400 °C, and this temperature was maintained for 90 min to wash out and reform light hydrocarbons. Subsequently, the heavy hydrocarbons prone to coking were burnt by introducing oxygen with the flow rate gradually increased from 5 to 40 mL min⁻¹ at 500 °C. Following this step, nitrogen was purged through the reactor for 130 min to displace residual oxygen. Then, a hydrogen stream at 10 mL min⁻¹ was introduced to reduce the oxidized nickel species back to metallic Ni. Finally, nitrogen was passed through the system until the bed temperature decreased to ambient conditions, yielding the regenerated Al₂-Ni_{INC}/NCNT catalyst.

Comment 1:

Catalytic results: The conversion of acetylene is given in rough numbers (100%), that means 99%, 99.99%? The industrial process requires less than 1 ppm in the final stream, although <10 ppm can be acceptable at the academic level. This has to be measured and explain how is measured (exact mass balance, not with a 0.5% error). A 10% of ethane is very much, the industrial process requires less than 2%, otherwise the economic penalty is huge.

Reply:

We sincerely thank you for your attention to this detail. The rough conversion of acetylene has been modified as the specific value (99.98% for Al₂-Ni_{INC}/NCNT catalyst and 99.93% for Ni_{INC}/NCNT catalyst). As accurately determined by online Gas Chromatography using an internal standard method, the initial concentration of acetylene was detected as 3109 ppm at 25 °C and the final acetylene concentration decreased to 0.8 ppm at 157 °C for the Al₂-Ni_{INC}/NCNT catalyst, which satisfies the industrial requirement (less than 1 ppm) in the final stream.

Based on the plot of acetylene conversion and selectivity as the function of temperature, it is found that the ethylene selectivity could be higher than 98% at lower

conversion (<75%). That is to say, the generated ethane is less than 2%, and thus possesses the high economy. This also complies with the operating at lower conversion in the industrial process to achieve high selectivity of the target product. Moreover, in this manuscript, we also emphasize higher conversion of acetylene, at this situation of which the ethylene selectivity could still be maintained at >90% to achieve ~90% of yield, which is reported as typical values for some Pd industrial catalysts (*J. Am. Chem. Soc.* 2018, 140, 8827; *III. US patents WO 03/106020, 2003, and US0137433, 2005*). However, it is still difficult to simultaneously achieve complete conversion of acetylene and >98% of ethylene selectivity at the industry.

Revisions made in the main text:

As expected, Al₂-Ni_{INC}/NCNT significantly enhances catalytic activity, achieving 99.98% conversion at 157 °C, far superior to Al₂/NCNT (49.75% @ 340 °C) and Ni_{INC}/NCNT (99.93% @ 200 °C) in Figure 3a and Figure S17-S18.

In contrast, the Ni_{INC} catalyst, which lacks adjacent Al dual-atom, exhibits markedly lower ethylene selectivity (43.70% at 99.93% conversion).

However, despite having the same adsorption configuration of reactants, the Al₂-Ni_{INC}/NCNT drives the reaction faster (99.98% conversion @157 °C).

Comment 2:

Catalyst nature: Why using CNTs if the support structure does not have any influence on the formation of the metal sites, as demonstrated by the Raman experiments? In this case, a simple N-doped carbon should be enough. Is N doping really essential? One could argue that O instead of N is stabilizing the dual sites (even perhaps Cl). The difference if XAS between N and O is very low. N XPS should help here. Any evidence that all the Cl is gone? It could still be on the catalyst.

The Al₁-catalyst is made by impregnation in water solution, and no characterization is given. The likely structure here is Al₂O₃ entities, thus comparison with the Al₂-catalyst is quite doubtful. In any case, any Al-N or Al-O single atom site is quite prone to decompose under H₂ atmospheres and severe heating, thus shedding more doubts on the catalyst stability under reaction conditions.

Figure 1b (PXRD) does not guarantee the formation of Ni aggregates, it looks the opposite.

Reply:

Thanks for your constructive question. Based on the referenced literature (*Dalton Trans.* 2022, 51, 10898), metallic Ni nanoclusters and Al dual-atom species typically exhibit no obvious Raman signals, primarily due to their high polarizability, which makes it difficult for atomic vibrations to induce a change in polarizability. In our situation, as shown in Figure S4, although the Raman spectra display similar patterns with the characteristic D and G peaks, along with comparable I_D/I_G intensity ratios across $Al_2-Ni_{INC}/NCNT$, $Al_2/NCNT$ and $Ni_{INC}/NCNT$ catalysts, it does not mean that the support structure has not influence on the formation of the metal sites due to the possibility of metal having no obvious Raman signals. To enhance the rigor of the manuscript, the relevant discussions have been revised accordingly.

Compared to a simple N-doped carbon, CNT is generally reported to have larger specific surface areas, which could facilitate the dispersion of active metal sites and thereby prevent agglomeration (*Nat. Catal.* 2023, 6, 818). Meanwhile, N-doped CNT could provide abundant nitrogen coordination sites for stabilizing active metal atoms (*Nat. Synth.* 2024, 3, 1427; *Nat. Chem.* 2020, 12, 764; *Nat. Catal.* 2025, 8, 248). Consequently, NCNT with high specific surface area is employed as the support for loading Ni_{INC} nanoclusters and anchoring the atomically dispersed Al_2 species.

Regarding the coordination environment of Al species, Cl ligands stabilizing the dual sites can be ruled out, as M-Cl coordination typically appears at higher radial distances, generally around 1.9 Å (*Angew. Chem. Int. Ed.* 2024, 63, e202401373). Furthermore, no Cl signal is detected in the XPS results of in $Al_2/NCNT$ and $Al_2-Ni_{INC}/NCNT$ catalysts (Figure S8), indicating complete removal of Cl species. With respect to Al-O coordination, Al K-edge X-ray absorption near-edge structure (XANES) spectra (Figure 2) could exhibit a higher oxidation state for Al species, particularly characteristic of Al_2O_3 . In contrast, the Al valence state of $Al_2-Ni/NCNT$ is lower, ranging from 0 to +3. Meanwhile, the N 1s XPS spectrum of $Al_2-Ni_{INC}/NCNT$ in Figure S9 exhibits a positive binding energy shift (~0.6 eV)

compared to that of NCNT (399.7 eV), suggesting the interaction of Al with N species. Taken together, these results indicate the absence of Al-O coordination and confirm that nitrogen atoms serve as the primary coordinating ligands stabilizing the Al dual sites.

The Al₁ catalyst was prepared via wet impregnation in an aqueous solution. No XRD diffraction signals attributable to Al₂O₃ are detected in Al₁/NCNT samples (Figure 1b). Furthermore, the aberration-corrected high-angle annular dark-field scanning transmission electron microscopy (AC HAADF-STEM) imaging reveals well-dispersed Al single-atom sites as isolated bright spots on the NCNT support in Al₁/NCNT, highlighted by white circles in Figure 1e. Fourier-transform extended X-ray absorption fine structure (FT-EXAFS) was also utilized to investigate the coordination environment of Al species (Figure 2c). A prominent peak for Al₁/NCNT appears at approximately 1.3 Å corresponding to Al-N coordination³³, while no Al-Al coordination appears. These above results suggest the generation of Al single atoms rather than Al₂O₃.

In Figure 1b, the intensity of XRD peaks associated with Ni nanoclusters is low. Two factors might account for this phenomenon: lower metal loading below the detection limit or smaller particle size of Ni species (*Science*, 2024, 385, 295). Meanwhile, to confirm the formation of Ni aggregates, we performed HAADF-STEM characterization over Al₂-Ni_{NC}/NCNT and Ni_{NC}/NCNT catalysts, as shown in Figure 1d and S5d. In both samples, Ni nanoclusters could be observed with an average particle size of approximately 2.5 nm.

Besides, additional and more comprehensive post-reaction characterizations have been provided in response to the reviewer's comments, including XRD, XPS, and XAS spectra, as well as regeneration testing. As shown in Figure S21-S23, XRD, XAS and XPS results reveal that the atomically-dispersed Al sites and Ni clusters could be well maintained under the thermal reaction conditions. Consequently, Al₂-Ni_{NC}/NCNT catalyst also exhibits good stability in a long-period running, even regeneration.

Figure 1 Aberration-corrected HAADF-STEM images of (c) Al₁/NCNT (d) Al₂-Ni_{NC}/NCNT (e) Al₁/NCNT without the circles.

Figure S5 Aberration-corrected HAADF-STEM images of (a) Al₂/NCNT (b) Al₂-Ni_{NC}/NCNT (c) Al₁/NCNT without the circles and (d) Ni_{NC}/NCNT.

Figure S8 Cl 2p XPS spectra of Al₂-Ni_{NC}/NCNT and Al₂/NCNT.

Figure S9 N 1s XPS spectra of $\text{Al}_2\text{-Ni}_{\text{NC}}/\text{NCNT}$ and NCNT.

Figure S21 XRD pattern of spent $\text{Al}_2\text{-Ni}_{\text{NC}}/\text{NCNT}$.

Figure S22 Fourier-transformed (a) Al and (b) Ni K-edge EXAFS spectra with fitting of spent

Al₂-Ni_{NC}/NCNT.

Figure S23 Al 2p XPS spectra of spent Al₂-Ni_{NC}/NCNT.

Figure 3h. Durability test on Al₂-Ni_{NC}/NCNT at 142 °C. (Reaction condition: a hydrogen to acetylene ratio of 20:1; space velocity = 9000 mL g⁻¹ h⁻¹; atmospheric pressure. Regeneration method is originated from the Jam domestic petrochemical for acetylene hydrogenation^{38,39}).

Revisions made in the main text:

In contrast, no diffraction signals attributable to Al species are detected in the Al₂/NCNT and Al₁/NCNT samples (Figure 1b).

Furthermore, aberration-corrected high-angle annular dark-field scanning transmission electron microscopy (AC HAADF-STEM) imaging reveals that the dispersed Al single-atom and dual-atom sites in Al₁/NCNT and Al₂/NCNT are clearly observed on the NCNT support, highlighted by white circles in Figure 1c, 1e and S5.

Based on this, Fourier-transform extended X-ray absorption fine structure (FT-EXAFS) was utilized to investigate the coordination environment of Al species

(Figure 2c). The two prominent peaks for Al₂-Ni_{INC}/NCNT and Al₂/NCNT appear at approximately 1.3 and 2.2 Å (without phase correction) corresponding to Al-N and Al-Al coordination, respectively³², whereas Al₁/NCNT exhibits only the former feature.

Notably, the support properties remain unchanged, as evidenced by Raman spectra (Figure S4), in which the spectra display similar patterns with the characteristic D peak at 1338 cm⁻¹ and G peak at 1562 cm⁻¹, along with comparable I_D/I_G intensity ratios across Al₂-Ni_{INC}/NCNT, Al₂/NCNT and Ni_{INC}/NCNT catalysts. This consistency indicates that the carbon structures exhibit similar levels of disorder or defects. Meanwhile, the metallic Ni nanoclusters and Al dual-atom species shows no obvious Raman signals primarily due to their high polarizability, which hinders atomic vibrations from inducing measurable changes in polarizability³¹.

Notably, the possibility of Al-Cl or Al-O coordination is excluded due to the higher peak position of M-Cl coordination at ~1.9 Å³³, and the evidence of Al interaction with N species from XPS results (Figure S8 and S9).

In addition to the diffraction features from the NCNT support, the X-ray diffraction (XRD) pattern of Al₂-Ni_{INC}/NCNT exhibits the weak characteristic diffraction peaks at 44.5° and 51.8° (PDF#04-0850, Figure 1b), similar to those observed in Ni_{INC}/NCNT. In contrast, no diffraction signals attributable to Al species are detected, in agreement with the pattern of atomically dispersed Al₂/NCNT and Al₁/NCNT samples (Figure 1b).

Following Ni deposition, as shown in Figure 1d and S5d, small nanoclusters with the lattice distances of 0.206 nm (corresponding to the Ni (111) plane) are observed, indicating a crystalline Ni structure with an average particle size of approximately 2.5 nm.

Notably, Al₂-Ni_{INC}/NCNT demonstrates exceptional thermal and chemical stability under hydrogenation conditions, maintaining both activity and selectivity for at least 150 h at 142 °C, with no visible decline even upon regeneration (Figure 3h). To investigate the nature and stability of this catalyst during reaction, XRD pattern and XAS analysis have been provided. As shown in Figure S21-S22, the

atomically-dispersed Al sites and Ni clusters of Al₂-Ni_{INC}/NCNT could be well maintained after the reaction. Meanwhile, Al 2p XPS results (Figure S23) further indicate that the average oxidation state of Al species well keeps between 0 and +3. That is to say, the atomically-dispersed Al sites and Ni clusters could be well maintained under the thermal reaction conditions, with excellent resistance to carbon deposition (Figure S24). In contrast, Ni_{INC}/NCNT shows a gradual decrease in ethylene selectivity, approximately with a loss of ~90% within the first five hours (Figure S25), and Raman measurements indicate severe coking (Figure S24).³⁹

Comment 3:

Reaction conditions: The reaction temperatures employed here (>150 °C, >300 °C for only Al) are extremely high for this reaction. The optimized reaction conditions employ 20:1 H₂: ethylene, this is much more than standard front-end conditions. Non-precious metal catalysts can perform the reactions under milder reaction conditions, see for instance *J. Am. Chem. Soc.* 2018, 140, 8827–8832 (this article should be cited and included in the comparative Table). This comparative Table is partially informative, it is true that hundreds of catalytic systems have been published, but the selection here is quite random, and the commercial catalyst (Pd-Al₂O₃) operates at much milder reaction conditions (35-70 °C) than indicated in the Table.

Reply:

We gratefully acknowledge the valuable comments. Based on the reported references (Tables S5 and S6), the atomically dispersed non-noble metal catalysts generally require a higher reaction temperature than industrial Pd-based catalysts (Tables S7), owing to the relatively weak hydrogen activation capability. In this work, the Al₁/NCNT and Al₂/NCNT catalysts operated at higher temperature are employed as reference samples. Upon construction of the synergistic Al₂-Ni_{INC}/NCNT catalyst, the reaction temperature could be effectively decreased, reaching a level comparable to that of most catalysts reported in current references, such as *J. Am. Chem. Soc.* 2018, 140, 8827. Inevitably, a trade-off between energy consumption and catalyst cost needs to be considered during industrial scaling-up. Although the reaction temperature

remains higher than that of commercial Pd catalysts, the development of non-noble Al₂-Ni_{NC}/NCNT catalyst in this work could significantly reduce the cost of catalysts.

The optimized H₂:C₂H₂ ratio of 20:1 was employed based on the reported references (*Nat Commun.* 2021, 12, 5770; *ACS Catal.* 2021, 11, 6073) and the industrial conditions (*Ethylene Industry* 2009, 21, 62), falling within the range of front-end condition of China Petroleum & Chemical Corporation (10-30% hydrogenation/0.5-1% acetylene). Meanwhile, based on your suggestion, the reference (*J. Am. Chem. Soc.* 2018, 140, 8827) has been added to the main text and Table S6, as well as the comparative data in the supplementary tables have been updated, which is systematically classified as Ni-based catalysts (Table S5), Cu-based and Fe-based catalysts (Table S6) as well as Pd-based catalysts (Table S7).

Table S5. Performance of Ni-based catalysts reported in references

Catalysts	Loading (%)	M amount (mg)	C ₂ H ₂ :H ₂ :C ₂ H ₄ (vol.%)	T (°C)	Conv. (%)	Sel. (%)
Al ₂ -Ni _{NC} /NCNT	0.28	0.56	1: 20: 99	157	99.98	90.26
Ni ₁ Cu ₂ /g-C ₃ N ₄ ¹⁰	3.1	4.65	0.5: 5: 25	170	100	90
NiGa ¹¹	10	5	1: 10: 20	180	70	75
NiCu/MMO ¹²	10	10	0.33:0.66:34.5	160	100	70
Ni-Cu/r-Al ₂ O ₃ ¹³	4.7	14.1	1: 10: 99	135	100	86
Ni@CeO ₂ ¹⁴	1.5	-	0.9: 1.9: 33.15	200	65	100
Ni ₃ ZnCo _{0.7} /OCNT ¹⁵	5	0.5	0.5: 4.5: 20	200	99	94
AgNi _{0.5} /SiO ₂ ¹⁶	1.09	0.327	1: 20: 79	220	99	65
CuNi _{0.125} /SiO ₂ ¹⁶	0.41	0.123	1: 20: 0	300	100	60
Ni/g-C ₃ N ₄ -T ¹⁷	1.31	0.655	1: 10: 20	175	100	84

NiCu/ZrO ₂ ¹⁸	4.6	1.15	0.5: 5: 25	170	98	86
NiCuFeGaGe/SiO ₂ ¹⁹	1	2	1: 10: 10	220	100	93
Ni _n /ND@G ²⁰	0.56	0.336	1: 10: 20	190	100	85
Na-Ni@CHA ²¹	3.5	7	0.5: 8: 5	180	100	97
Ni ₁ MoS/Al ₂ O ₃ ²²	3.5	17.5	0.15: 3.03:15	125	100	90
Ni ₁ Sb ₂ ²³	10.7	32.1	0.5: 2.5: 30	260	100	93

Table S6. Performance of Cu-based and Fe-based catalysts reported in references

Catalysts	loading (wt%)	M amount (mg)	C ₂ H ₂ :H ₂ :C ₂ H ₄ (vol.%)	T (°C)	Conv. (%)	Sel. (%)
CuZn/NC ²⁴	1.15	-	0.33:0.66:33	180	97	97.5
Cu ₁ /C ₃ N ₄ ¹⁰	8.1	12.15	0.5: 5: 25	200	<10	<60
Cu ₁ /ND@G ²⁵	0.5	0.5	1: 10: 20	200	100	93
Cu/Al ₂ O ₃ ²⁶	0.5	4	1: 10:50	188	100	91
Cu ₁ /ND@G ²⁷	0.25	0.5	1: 10: 20	200	95	98
Cu/Fe _{0.16} MgO _{x28}	8.29	16.58	0.33: 1.02: 32.86	215	100	95
CuPd _{0.006} /SiO ₂ ²⁹	4.96	1.488	1:20:20	160	100	85
CuBi/SiO ₂ (HTR) ³⁰	-	-	0.5:10:20	100	100	91.1
Cu(OH) ₂ ³¹	-	-	0.4:10:88.88	110	100	51
Cu ₂ O ³²	-	-	-	190	100	50.8

Cu/ZrO ₂ ³³	8.5	2.125	1:10:20	220	15	100
Fe-MOF ³⁴	2.82	0.705	1.2 ^a	150	>99.917	>90

^a acetylene concentration in an ethylene (1 mL/min, 2 bar) and H₂ (2 mL/min, 4 bar) flow

Table S7. Performance of Pd-based catalysts reported in references

Catalysts	Loading (%)	M amount (mg)	C ₂ H ₂ :H ₂ :C ₂ H ₄ (vol.%)	T (°C)	Conv. (%)	Sel. (%)
Pd/Al ₂ O ₃ ³⁵	2.58	1.032	1: 2: 99	138	100	88
Pd ₁ Cu ₁ /ND@G ³⁶	0.09	0.027	1: 10: 20	110	100	92
Pd ₁ /TiO ₂ ³⁷	0.146	0.0219	1: 10: 20	120	100	>50
Pd ₁ Au ₁ @ ³⁸	2.9	0.87	1.1: 9.1: 89.8	100	100	89
Pd ₁ @Cu-SiW ³⁹	0.41	1.23	0.5: 5: 50	120	100	93
Pd ₈ Zn ₄₄ ⁴⁰	7.8	1.17	1: 18: 31	160	100	90
Pd/Bi ₂ O ₃ /TiO ₂ ⁴¹	2.5	0.75	1: 20: 20	44	90	91
Pd ₁ /N-graphene ⁴²	2.3	1.15	1: 20: 20	125	99	94
B2 CuPd ⁴³	9.5	1.9	0.5: 5: 10	90	100	95
Pd ₁ /ND@G ⁴⁴	0.11	0.033	1:10:20	180	100	90
PdZn-1.2@ZIF-8C ⁴⁵	0.1	0.1	0.65:5:50	120	85	80
PdIn/MgAl ₂ O ₄ ⁴⁶	2	0.5	0.5:5:50	90	96	92

Pd@SOD ⁴⁷	0.099	0.099	1:10:0	150	99.8	94.5
Pd@C/CNT ⁴⁸	1.5	0.075	0.5:3:20	150	93	70
Pd ₂ Sn/C ⁴⁹	0.97	2.716	1:2:99	160	97.5	91
Pd ₄ S/CNTs ⁵⁰	1	0.5	1:1.8:9	200	100	85
PdAg/TiO ₂ ⁵¹	0.64	0.64	1:5:20	95	99	60
Pd-Fe ₃ O ₄ -H ⁵²	-	-	1:6:0	85	100	80
ISA-Pd/MPNC ⁵³	0.0433	0.0866	0.5:5:50	120	100	80
Pd-SAs @ZIF-8 ⁵⁴	0.16	0.00256	0.5:5:50	120	96	93.4
Pd ₁ /C ₃ N ₄ ⁵⁵	0.5	-	0.5:1:25	115	99	83
Pd/Ni(OH) ₂ ⁵⁶	0.005	-	0.65:5:50	105	80	70
Pd/MgO ⁵⁷	0.05	0.05	1:2:0	200	100	82
Pd/MgO ⁵⁸	0.16	0.032	1:10:20	140	100	70
Pd ₁ /ZnO ⁵⁹	0.01	50	2:20:40	100	100	80
Pd ₁ /CeO ₂ ⁶⁰	0.098	-	2:20:40	160	100	85
Pd/MgAl ₂ O ₄ ⁶¹	0.1	0.01	1:5:20	120	96	87
Pd ₁ -N ₈ /CNT ⁶²	0.0092	0.0046	2:4:50	40	83	98
Pd ₃ /GDY ⁶³	0.4	-	0.33:0.66:32.8	160	100	97
PdZn/ZnO ⁶⁴	0.7	0.7	2:20:40	80	92	89
PdAg/Al ₂ O ₃ ⁶⁵	0.0003	-	0.738:0.738:64.594	45	90	80

References

- Gu J, Jian M, Huang L, et al. Synergizing metal-support interactions and spatial confinement boosts dynamics of atomic nickel for hydrogenations. *Nat. Nanotechnol.* **2021**, 16, 1141-1149.
- Cao Y, Zhang H, Ji S, et al. Adsorption site regulation to guide atomic design of Ni-Ga catalysts for acetylene semi-hydrogenation. *Angew. Chem. Int. Ed.* **2020**, 59, 11647-11652.
- Liu Y, Zhao J, Feng J, et al. Layered double hydroxide-derived Ni-Cu nanoalloy catalysts for semi-hydrogenation of alkynes: improvement of selectivity and anti-coking ability via alloying of Ni and Cu. *J. Catal.* **2018**, 359, 251-260.
- Song Y, Weng S, Xue F, et al. Understanding the Role of Coordinatively Unsaturated Al³⁺ Sites on Nanoshaped Al₂O₃ for Creating Uniform Ni-Cu Alloys for Selective Hydrogenation of Acetylene. *ACS Catal.* **2023**, 13, 1952-1963.

14. Riley C, Zhou S, Kunwar D, et al. Design of effective catalysts for selective alkyne hydrogenation by doping of ceria with a single-atom promotor. *J. Am. Chem. Soc.* **2018**, 140, 12964-12973.
15. Niu Y, Huang X, Wang Y, et al. Manipulating interstitial carbon atoms in the nickel octahedral site for highly efficient hydrogenation of alkyne. *Nat. Commun.* **2020**, 11, 1-9.
16. Liu H, Chai M, Pei G, et al. Effect of IB-metal on Ni/SiO₂ catalyst for selective hydrogenation of acetylene. *Chin. J. Catal.* **2020**, 41, 1099-1108.
17. Zhou H, Li B, Fu H, et al. Sulfur-doped g-C₃N₄-supported Ni species with a wide temperature window for acetylene semi-hydrogenation. *ACS Sustain. Chem. Eng.* **2022**, 10, 4849-4861.
18. Li Z, Zhang J, Tian J, et al. Unveiling the origin of enhanced catalytic performance of NiCu alloy for semi-hydrogenation of acetylene. *Chem. Eng. J.* **2022**, 450, 138244.
19. Ma J, Xing F, Nakaya Y, et al. Nickel-based high-entropy intermetallic as a highly active and selective catalyst for acetylene semihydrogenation. *Angew. Chem. Int. Ed.* **2022**, 134, e202200889.
20. Sui C, Ma H, Huang F, et al. Fully Exposed Nickel Clusters for Semihydrogenation of Acetylene. *ACS Catal.* **2024**, 14, 14689-14695.
21. Chai Y, Wu G, Liu X, et al. Acetylene-selective hydrogenation catalyzed by cationic nickel confined in zeolite. *J. Am. Chem. Soc.* **2019**, 141, 9920-9927.
22. Fu B, McCue A J, Liu Y, et al. Highly selective and stable isolated non-noble metal atom catalysts for selective hydrogenation of acetylene. *ACS Catal.* **2021**, 12, 607-615.
23. Ge X, Dou M, Cao Y, et al. Mechanism driven design of trimer Ni₁Sb₂ site delivering superior hydrogenation selectivity to ethylene. *Nat. Commun.* **2022**, 13, 5534.
24. Yue Y, Wang B, Jin C, et al. Tailoring Cu-Zn dual-atom sites with reordering d-orbital splitting manner for highly efficient acetylene semihydrogenation. *ACS Catal.* **2024**, 14, 3900-3911.
25. Huang F, Peng M, Chen Y L, et al. Insight into the activity of atomically dispersed Cu catalysts for semihydrogenation of acetylene: impact of coordination environments. *ACS Catal.* **2021**, 12, 48-57.
26. Shi X X, Lin Y, Huang L, et al. Copper catalysts in semihydrogenation of acetylene: from single atoms to nanoparticles. *ACS Catal.* **2020**, 10, 3495-3504.

27. Huang F, Deng Y C, Chen Y L, et al. Anchoring Cu₁ species over nanodiamond-graphene for semi-hydrogenation of acetylene. *Nat. Commun.* **2019**, 10, 4431.
28. Fu F, Liu Y, Li Y, et al. Interfacial bifunctional effect promoted non-noble Cu/Fe_yMgO_x catalysts for selective hydrogenation of acetylene. *ACS Catal.* **2021**, 11, 11117-11128.
29. Pei G, Liu X, Yang X, et al. Performance of Cu-alloyed Pd single-atom catalyst for semihydrogenation of acetylene under simulated front-end conditions. *ACS Catal.* **2017**, 1491-1500.
30. Zhou S, Zeng A, Lu C, et al. Bi-modified Cu-based catalysts for acetylene hydrogenation: leveraging dispersion and hydrogen spillover. *Inorg. Chem.* **2024**, 63, 11802-11811.
31. Lu C, Zeng A, Wang Y, et al. Copper-based catalysts for selective hydrogenation of acetylene derived from Cu(OH)₂. *ACS omega.* **2021**, 6, 3363-3371.
32. Liu T, Xiong J, Luo Q, et al. Hydrophobic surface modification of Cu-based catalysts for enhanced semihydrogenation of acetylene in excess ethylene. *ACS Catal.* **2024**, 14, 5838-5846.
33. Li Z, Zhang J, Tian J, et al. Unveiling the origin of enhanced catalytic performance of NiCu alloy for semi-hydrogenation of acetylene. *Chem. Eng. J.* **2022**, 450, 138244.
34. Tejada-Serrano M, Mon M, Ross B, et al. Isolated Fe (III)-O sites catalyze the hydrogenation of acetylene in ethylene flows under front-end industrial conditions. *J. Am. Chem. Soc.* **2018**, 140, 8827-8832.
35. Xue F, Li Q, Lv M, et al. Atomic three-dimensional investigations of Pd nanocatalysts for acetylene semi-hydrogenation. *J. Am. Chem. Soc.* **2023**, 145, 26728-26735.
36. Huang F, Peng M, Chen Y, et al. Low-temperature acetylene semi-hydrogenation over the Pd₁-Cu₁ dual-atom catalyst. *J. Am. Chem. Soc.* **2022**, 144, 18485-18493.
37. Guo Y, Huang Y, Zeng B, et al. Photo-thermo semi-hydrogenation of acetylene on Pd₁/TiO₂ single-atom catalyst. *Nat. Commun.* **2022**, 13, 2648.
38. Ballesteros-Soberanas J, Martín N, Bacic M, et al. A MOF-supported Pd₁-Au₁ dimer catalyses the semihydrogenation reaction of acetylene in ethylene with a nearly barrierless activation energy. *Nat Catal.* **2024**, 7, 452-463.
39. Liu Y, Wang B, Fu Q, et al. Polyoxometalate-based metal-organic framework as molecular sieve for highly selective semi-hydrogenation of acetylene on isolated single Pd atom sites. *Angew. Chem. Int. Ed.* **2021**, 60, 22522-22528.

40. Dasgupta A, He H, Gong R, et al. Atomic control of active-site ensembles in ordered alloys to enhance hydrogenation selectivity. *Nat. Chem.* **2022**, 14, 523-529.
41. Zou S, Lou B, Yang K, et al. Grafting nanometer metal/oxide interface towards enhanced low-temperature acetylene semi-hydrogenation. *Nat. Commun.* **2021**, 12, 5770.
42. Zhou S, Shang L, Zhao Y, et al. Pd single-atom catalysts on nitrogen-doped graphene for the highly selective photothermal hydrogenation of acetylene to ethylene. *Adv. Mater.* **2019**, 31, 1900509.
43. Gao Q, Yan Z, Zhang W J, et al. Atomic layers of B₂ CuPd on Cu nanocubes as catalysts for selective hydrogenation. *J. Am. Chem. Soc.* **2023**, 145, 36, 19961-19968.
44. Huang F, Deng Y, Chen Y, et al. Atomically dispersed Pd on nanodiamond/graphene hybrid for selective hydrogenation of acetylene. *J. Am. Chem. Soc.* **2018**, 140, 13142-13146.
45. Li J, Huang H, Li Y, et al. Stable and size-controllable ultrafine Pt nanoparticles derived from a MOF-based single metal ion trap for efficient electrocatalytic hydrogen evolution. *J. Mater. Chem. A.* **2019**, 7, 20239-20246.
46. Feng Q, Zhao S, Wang Y, et al. Isolated single-atom Pd sites in intermetallic nanostructures: high catalytic selectivity for semihydrogenation of alkynes. *J. Am. Chem. Soc.* **2017**, 139, 7294-7301.
47. Wang S, Zhao Z J, Chang X, et al. Activation and spillover of hydrogen on sub-1 nm palladium nano-clusters confined within sodalite zeolite for the semi-hydrogenation of alkynes. *Angew. Chem. Int. Ed.* **2019**, 58, 7668-7672.
48. Zhang L, Ding Y, Wu K H, et al. Pd@C core-shell nanoparticles on carbon nanotubes as highly stable and selective catalysts for hydrogenation of acetylene to ethylene. *Nanoscale.* **2017**, 9, 14317-14321.
49. Li R, Yue Y, Chen Z, et al. Selective hydrogenation of acetylene over Pd-Sn catalyst: Identification of Pd₂Sn intermetallic alloy and crystal plane-dependent performance. *Appl. Catal. B Environ.* **2020**, 279, 119348.
50. McCue A J, Guerrero-Ruiz A, Rodríguez-Ramos I, et al. Palladium sulphide-A highly selective catalyst for the gas phase hydrogenation of alkynes to alkenes. *J. Catal.* **2016**, 340, 10-16.
51. Riyapan S, Zhang Y, Wongkaew A, et al. Preparation of improved Ag-Pd/TiO₂ catalysts using

the combined strong electrostatic adsorption and electroless deposition methods for the selective hydrogenation of acetylene. *Catal. Sci. Technol.* **2016**, 6, 5608-5617.

52. Wu P, Tan S, Moon J, et al. Harnessing strong metal-support interactions via a reverse route. *Nat. Commun.* **2020**, 11, 3042.

53. Feng Q, Zhao S, Xu Q, et al. Mesoporous nitrogen-doped carbon-nanosphere-supported isolated single-atom Pd catalyst for highly efficient semihydrogenation of acetylene. *Adv. Mater.* **2019**, 31, 1901024.

54. Wei S, Li A, Liu J C, et al. Direct observation of noble metal nanoparticles transforming to thermally stable single atoms. *Nat. Nanotechnol.* **2018**, 13, 856-861.

55. Huang X, Xia Y, Cao Y, et al. Enhancing both selectivity and coking-resistance of a single-atom Pd₁/C₃N₄ catalyst for acetylene hydrogenation. *Nano Res.* **2017**, 10, 1302-1312.

56. Hu M, Zhang J, Zhu W, et al. 50 ppm of Pd dispersed on Ni(OH)₂ nanosheets catalyzing semi-hydrogenation of acetylene with high activity and selectivity. *Nano Res.* **2018**, 11, 905-912.

57. Tao X, Nan B, Li Y, et al. Highly active isolated single-atom Pd catalyst supported on layered MgO for semihydrogenation of acetylene. *ACS Appl. Energy Mater.* **2022**, 5, 10385-10390.

58. Guo Y, Qi H, Su Y, et al. High Performance of Single-atom Catalyst Pd₁/MgO for Semi-hydrogenation of Acetylene to Ethylene in Excess Ethylen. *ChemNanoMat.* **2021**, 7, 526-529.

59. Zhou H, Yang X, Wang A, et al. Pd/ZnO catalysts with different origins for high chemoselectivity in acetylene semi-hydrogenation. *Chin. J. Catal.* **2016**, 37, 692-699.

60. Guo Y, Li Y, Du X, et al. Pd single-atom catalysts derived from strong metal-support interaction for selective hydrogenation of acetylene. *Nano Res.* **2022**, 15, 10037-10043.

61. Li Z, Lin G, Chen Y, et al. Regulating metal-support interactions of Pd/MgAl₂O₄ for efficient selective hydrogenation of acetylene. *Catal. Today.* **2023**, 423, 114253.

62. Hu M, Wu Z, Yao Z, et al. N₈ stabilized single-atom Pd for highly selective hydrogenation of acetylene. *J. Catal.* **2021**, 395, 46-53.

63. Li R, Yue Y, Chen X, et al. Graphdiyne anchoring to construct highly dense palladium trimer active sites for the selective hydrogenation of acetylene. *Nano Res.* **2023**, 16, 6167-6177.

64. Zhou H, Yang X, Li L, et al. PdZn intermetallic nanostructure with Pd-Zn-Pd ensembles for highly active and chemoselective semi-hydrogenation of acetylene. *ACS Catal.* **2016**, 6,

1054-1061.

65. Dehghani O, Rahimpour M and Shariati A. An experimental approach on industrial Pd-Ag supported α -Al₂O₃ catalyst used in acetylene hydrogenation process: mechanism, kinetic and catalyst decay. *Processes*. **2019**, 7, 136.

Comment 4:

Reaction mechanism: The characterization (HRTEM) claims that the dual Al sites are nearby the Ni clusters, surrounding them. There is not clear evidence of this (EDS mapping, Al-Ni reduced distances in XAFS, the XPS shift is very doubtful). In any case, H-spillover from Ni to Al sites would not require any spatial approximation, and that has not been considered. Please check (a simple test with WO₃ will confirm the H spillover).

Reply:

We gratefully acknowledge the reviewer's insightful comments. As shown in EDS images, the Ni clusters are surrounded by the Al atoms.

Figure S6 The EDX element maps of Al₂-Ni_{NC}/NCNT catalyst

Thanks for your constructive comment. We have conducted the hydrogen spillover test using WO₃ as the indicator material. In detail, WO₃ is stuck to Al₂-Ni_{NC}/NCNT using the reported method from the reference (*Angew. Chem. Int. Ed.* 2024, 63, e202316319). As shown in Figure S28 below, the mixture of WO₃ and Al₂-Ni_{NC}/NCNT exhibits a distinct color change from canary yellow to dark blue after reaction, which confirms the occurrence of hydrogen spillover.

Figure S28. Color change of WO_3 and $\text{Al}_2\text{-Ni}_{\text{NC}}/\text{NCNT}$ mixtures

Revisions made in the main text:

Additionally, it is observed that while the Al distribution remains uniform, Ni nanoclusters are segregated and surrounded by abundant Al dual-atom in $\text{Al}_2\text{-Ni}_{\text{NC}}/\text{NCNT}$, as confirmed by the HAADF-STEM and EDS image (Figure 1d and S6).

Meanwhile, an experiment was conducted in which WO_3 was mixed with $\text{Al}_2\text{-Ni}_{\text{NC}}/\text{NCNT}$, and a distinct color change from canary yellow to dark blue was observed (Figure S28), confirming the occurrence of hydrogen spillover⁴⁵.

Comment 5:

Minor comment: The meaning of TPSR should be specified (not said neither in the main text or SI).

Reply:

Thanks very much for your attention to this detail. TPSR stands for temperature-programmed surface reaction, which has been specified in the main text.

Revisions made in the main text:

As a result, and as expected, ethane and oligomers are prominently detected in the $\text{Ni}_{\text{NC}}/\text{NCNT}$ catalyst, with no ethylene formation. This observation aligns well with temperature-programmed surface reaction (TPSR) results (Figure S29), which show that acetylene hydrogenation on $\text{Al}_2\text{-Ni}_{\text{NC}}/\text{NCNT}$ starts at a lower temperature (77 °C).

Revisions made in the Supporting Information:

Temperature-programmed surface reaction (TPSR) experiment was performed in a reaction cell. 0.2 g of sample was pretreated in N_2 for 30 min. Then, the gas was

switched to 0.31% C₂H₂/30.40% C₂H₄/6.20% H₂ (balanced with nitrogen) with flow rate of 30 mL/min and heated from 40 to 380 °C for data recording. The products (m/z of C₂H₂, C₂H₄ and C₂H₆ being 26, 27 and 15, respectively) were measured via mass spectrometer.

Comment 6:

In conclusion, although recognizing that the activation of acetylene on atomically-dispersed Al sites for the semi-hydrogenation reaction is of academic interest, the study here does not demonstrate that this is occurring, since the stability of the Al site during reaction and a significant catalytic performance beyond that of the Ni sites, is not demonstrated.

Reply:

Thanks for your constructive comment. As suggested in your comments, we have conducted additional experiments including HAADF-STEM and XAS characterizations of the Al₂-Ni_{INC}/NCNT catalyst after reaction, hydrogen spillover and so on to improve this manuscript.

Response to reviewers

Reviewer #1

I have carefully reviewed the revised manuscript and the authors' detailed responses to my comments. They have made substantial and satisfactory efforts in addressing all the raised issues. The manuscript has been significantly improved and is now clear, compelling, and ready for publication. Therefore, I am pleased to recommend its acceptance in its current form.

Reply:

We are deeply grateful for your positive assessment and confirmation that our revised manuscript is acceptable for publication.

Reviewer #2

The authors have revised the manuscript and have addressed my concerns adequately.

Reply:

We are deeply grateful for your positive assessment.

Reviewer #3

I recognize that the authors have made an effort to answer to all Reviewer's requirements, including mine. However, two main issues still puzzle me. Regarding characterization of the spent catalyst, the only technique shown here that gives real information about the isolated Al atoms is the XAS, but the spectra are presented in a very general way, without comparing with the fresh one. This has to be done.

Reply:

Thanks for your valuable comments and critical questions. Based on the suggestion of reviewer, XAS results of fresh and spent Al₂-Ni_{INC}/NCNT have been compared in Figure S22, in which both exhibit the similar coordination features, indicating that the atomically-dispersed Al sites and Ni clusters could be well maintained under the thermal reaction conditions.

Figure S22 Fourier-transformed (a) Al and (b) Ni K-edge EXAFS spectra of spent $\text{Al}_2\text{-Ni}_{\text{NC}}/\text{NCNT}$ compared with fresh ones (grey: C; blue: N; pink: Al; blue grey: Ni)

Second, and more important, one of the Reviewers insightfully asked about the regeneration of the solid catalyst with O_2 , to burn the residual carbonaceous molecules. The authors claim that the isolated Al sites are able to bear the treatment with O_2 at 500°C without oxidation. This is completely unbelievable, if so, this is the discovery here: A solid that keeps Al unoxidized at 500°C in O_2 . Catalysis with O_2 should be tested! Trying to be positive here, the only explanation that comes to my mind in order to accept that the catalyst preserves structure and activity after treatment with O_2 at 500°C , is that the Al sites are originally oxidized and that, under the H_2 conditions and with the Ni sites continuously supplying H atoms by spillover during reaction, the Al sites become active for acetylene. However, the most plausible explanation is that Al has nothing to do with acetylene activation, but exerts some kind of modification on Ni. All these arguments should be considered by the authors before publication.

Reply:

Sincerely apologies for the ambiguous description that led to the misinterpretation. In the previous response and revised manuscript, we described the spent catalysts as below:

“ $\text{Al}_2\text{-Ni}_{\text{NC}}/\text{NCNT}$ demonstrates exceptional thermal and chemical stability under hydrogenation conditions, maintaining both activity and selectivity for at least 150 h at 142°C , with no visible decline even upon regeneration (Figure 3h). To investigate the

nature and stability of this catalyst during reaction, XRD pattern and XAS analysis have been provided. As shown in Figure S21-S22, the atomically-dispersed Al sites and Ni clusters of Al₂-Ni_{NC}/NCNT could be maintained after the reaction. Meanwhile, Al 2p XPS results (Figure S23) further indicate that the average oxidation state of Al species well keeps between 0 and +3, which is basically in agreement with the fresh one. Additionally, the excellent resistance to carbon deposition is also confirmed (Figure S24)”.

“Raman and XAS results in the figure below demonstrate that the industrial regeneration process could effectively remove the oligomers and coking from the surface of Al₂-Ni_{NC}/NCNT catalyst, while the atomically dispersed Al sites and Ni nanoclusters remain intact. This is further supported by the catalytic performance of the regenerated Al₂-Ni_{NC}/NCNT catalyst, which is basically consistent with that of the corresponding fresh catalyst”.

Notably, the industrial regeneration process not only introduces oxygen to remove heavy hydrocarbons as the precursor of coking, but also employs subsequent nitrogen and hydrogen purging to eliminate residual oxygen and fully reduce the oxidized active species back to their fresh states. These could be confirmed by XAS and XPS spectra (Figure S22 and S23). Specifically, the active Al species in the fresh catalyst exhibit a valence state intermediate between 0 and +3, which can be deeply oxidized to higher oxidation states (e.g., inert Al³⁺) during regeneration. Upon subsequent exposure to H₂, these inert Al species are reduced and reactivated, forming sites capable of acetylene adsorption, which could be supported by both spectroscopic evidence and DFT calculations (Figures 4-5 and S35-S39).

Figure S22 Fourier-transformed (a) Al and (b) Ni K-edge EXAFS spectra of spent $\text{Al}_2\text{-Ni}_{\text{NC}}/\text{NCNT}$ compared with fresh ones (grey: C; blue: N; pink: Al; blue grey: Ni)

Figure S23 Al 2p XPS spectra of spent $\text{Al}_2\text{-Ni}_{\text{NC}}/\text{NCNT}$

To make it better understand for readers, the corresponding discussions have been modified as below.

Revisions made in the main text:

The regeneration method originates from Jam domestic petrochemical's process for acetylene hydrogenation and primarily involves coke combustion through the introduction of oxygen at 500 °C, followed by a hydrogen treatment step to reduce the oxidized Al and Ni species back to their fresh states^{37,38}.

Notably, $\text{Al}_2\text{-Ni}_{\text{NC}}/\text{NCNT}$ demonstrates exceptional thermal and chemical stability under hydrogenation conditions, maintaining both activity and selectivity for at least 150 h at 142 °C, with no visible decline even upon regeneration (Figure 3h). To

investigate the nature and stability of this catalyst during reaction, XRD pattern and XAS analysis have been provided. As shown in Figure S21-S22, the atomically-dispersed Al sites and Ni clusters of Al₂-Ni_{INC}/NCNT could be maintained after the reaction; meanwhile, Al 2p XPS results (Figure S23) further indicate that the average oxidation state of Al species well keeps between 0 and +3, following the coking combustion and a subsequent hydrogen activation, which is basically in agreement with the fresh one. Additionally, the excellent resistance to carbon deposition is also confirmed (Figure S24).

Based on the results from *in situ* spectroscopic characterizations and theoretical calculations, it could be regarded that the construction of the synergistic Ni nanoclusters and Al₂ active centers offers excellent activity and selectivity, in which nickel species are responsible for hydrogen dissociation, while the Al₂ sites trigger the spillover of active H* to react with π -bound acetylene and weaken the desorption of ethylene instead of simply modify Ni, inhibiting over-hydrogenation and polymerization.

Revision made in the Supporting Information:

The used Al₂-Ni_{INC}/NCNT catalyst by oxidative treatment to remove oligomers, following a protocol adapted from industrial practices used at Jam domestic petrochemical for acetylene hydrogenation¹. Firstly, a nitrogen stream at a flow rate of 100 mL min⁻¹ was fed to the reactor at 115 °C for 60 min. The temperature was then increased to 165 °C while maintaining the nitrogen flow to stabilize the catalyst bed. In the next step, in addition to nitrogen, water vapor (H₂O (g)) at a flow rate of 0.02 mL min⁻¹ was injected as the temperature was raised from 165 to 400 °C, and this temperature was maintained for 90 min to wash out and reform light hydrocarbons. Subsequently, the heavy hydrocarbons prone to coking were burnt by introducing oxygen with the flow rate gradually increased from 5 to 40 mL min⁻¹ at 500 °C. Subsequently, the heavy hydrocarbons prone to coking were burnt by introducing oxygen with the flow rate gradually increased from 5 to 40 mL min⁻¹ at 500 °C. Following this step, nitrogen was purged through the reactor for 130 min to displace residual oxygen. Then, a subsequent hydrogen stream at 10 mL min⁻¹ was introduced to reduce the oxidized Al and Ni species back to their fresh states. Finally, nitrogen was

passed through the system until the bed temperature decreased to ambient conditions, yielding the regenerated Al₂-Ni_{NC}/NCNT catalyst.